# On the Interplay of Pre-Training, Mid-Training, and RL on Reasoning Language Models

**Charlie Zhang** [1]  **Graham Neubig** [1]  **Xiang Yue** [1][†]

## Abstract

Recent reinforcement learning (RL) techniques have yielded impressive reasoning improvements in language models, yet it remains unclear whether RL truly extends a model's reasoning ability beyond pre-training. A central challenge is the lack of control in modern training pipelines, where opaque pre-training data, underexplored mid-training, and complex RL interactions obscure causal effects. To resolve this ambiguity, we develop a controlled experimental framework that isolates the causal contributions of pre-training, mid-training, and RL-based post-training. Our approach employs synthetic reasoning tasks with explicit atomic operations, parseable step-by-step reasoning traces, and systematic manipulation of training distributions. We evaluate along: *extrapolative generalization* to more complex compositions and *contextual generalization* across surface contexts. Using this framework, we reconcile competing views on RL's effectiveness: 1) RL produces true capability gains (pass@128) only when pre-training leaves sufficient headroom and RL data target the model's *edge of competence*, tasks that are difficult but not yet out of reach. 2) Contextual generalization requires minimal yet sufficient pre-training exposure, after which RL reliably transfers. 3) Mid-training significantly enhances performance under fixed compute compared with RL alone, demonstrating its central but underexplored role. 4) Process-level rewards reduce reward hacking and improve reasoning fidelity. Together, these results clarify the interplay between pre-training, mid-training, and RL, offering a foundation for improving reasoning language models training strategies. Codes and data are avaialble at Github and HuggingFace

[1]Language Technologies Institute, Carnegie Mellon University, Pittsburgh, PA, USA. Correspondence to: Xiang Yue <xiangyue.work@gmail.com>.

*Proceedings of the 43rd International Conference on Machine Learning*, Seoul, South Korea. PMLR 306, 2026. Copyright 2026 by the author(s).

## 1. Introduction

Reinforcement learning (RL) has recently improved the reasoning capabilities of language models (LMs) (DeepSeek-AI et al., 2025; OpenAI et al., 2024). Yet a fundamental question remains unresolved: *does post-training truly extend a model's reasoning ability beyond what is acquired during pre-training*? The literature offers conflicting views: some work characterizes RL as a capability refiner (Yue et al., 2025; Wu et al., 2025; Shao et al., 2025; Yeo et al., 2025), while others present evidence of substantial reasoning gains beyond pre-training (Wen et al., 2025; Yuan et al., 2025; Sun et al., 2025a).

A major source of this discrepancy is that prior analyses rely on *uncontrolled* training environments. Modern LMs are pre-trained on massive, opaque internet corpora whose composition is fundamentally unknown. As a result, we cannot ascertain which reasoning primitives the base model has already internalized. Consequently, this lack of control makes it challenging to isolate the causal effect of post-training and to understand how pre-training and post-training jointly shape reasoning behavior.

Meanwhile, an additional stage, *mid-training*,[1] has recently emerged as a key component of modern LM pipelines (Wang et al., 2025; Liu et al., 2025a). Mid-training acts as an intermediate distributional bridge between broad pre-training corpora and specialized post-training objectives, expanding the model's primitive coverage and aligning its internal representations with the tasks emphasized during RL, which has become increasingly central to the debate: it may explain why RL sometimes produces striking generalization improvements, yet fails in other settings (Wang et al., 2025). This motivates the core question of our work: **What is the interplay between pre-training, mid-training, and RL in shaping the reasoning capabilities of LMs?**

The goal of this work is to convincingly answer this question in a controlled manner, following previous work in this vein (Ye et al., 2024; Zhou et al., 2025b). Specifically, we perform controlled experiments to disentangle how pre-training, mid-training, and RL-based post-training individually and jointly influence reasoning generalization.

---

[1]Some literature call this stage continued pre-training (CPT).

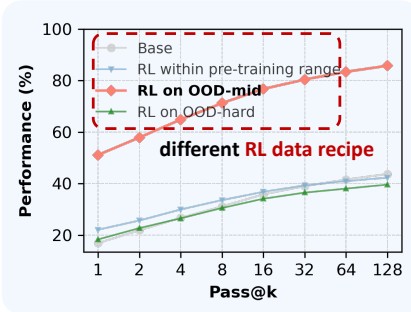 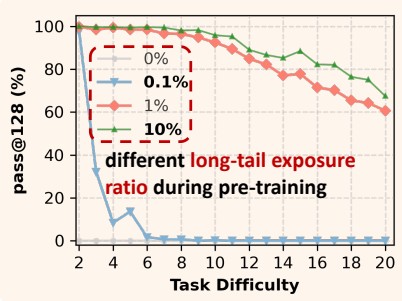 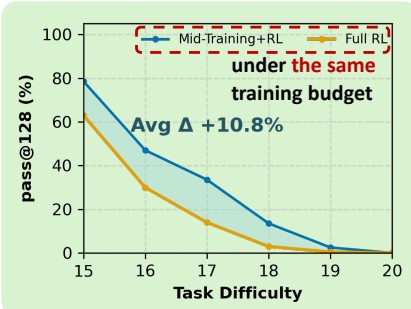

*Figure 1.* **Interplay of pre-, mid-, and post-training in LM reasoning. Left:** RL yields genuine extrapolative gains only when task difficulty slightly exceeds the pre-training range; gains vanish when tasks are already covered or too out-of-distribution (up to +42% pass@128 when well-calibrated). **Mid:** Contextual generalization requires minimal yet sufficient pre-training exposure to long-tail contexts. RL fails with near-zero exposure but generalizes robustly with sparse exposure ($\geq$1%), yielding up to +60% pass@128. **Right:** A mid-training stage bridging pre-training and RL substantially improves OOD reasoning under fixed compute, with mid-training + RL outperforming RL alone by +10.8% on OOD-hard tasks.

To this end, we build a fully controlled framework that isolates the contributions of each training stage. Our design is based on three principles: (i) *fully controllable synthetic reasoning tasks* with explicit atomic operations and DAG-defined dependency structure; (ii) *observable, parseable reasoning processes* enabling process-level evaluation and reducing reward or evaluation hacking; and (iii) *systematic manipulation* of pre-/mid-/post-training distributions to attribute causal effects to each stage.

We evaluate reasoning along two dimensions: 1) *Extrapolative (Depth) generalization* asks whether models can solve problems *more complex* than those seen in pre-training by composing learned primitives into deeper structures. 2) *Contextual (Breadth) generalization* asks whether models can *transfer* reasoning skills across novel surface contexts with equivalent underlying logic. Together, these axes capture compositional and transfer reasoning abilities relevant to real-world LMs. Scaling and real-world benchmark experiments further validate that these conclusions hold beyond the main controlled setting. Using this framework, we uncover several insights into how the three training stages interact.

**Firstly**, the two competing views on whether RL genuinely improves a base model's reasoning ability do not truly conflict. RL produces true capability gains only when two conditions hold: (i) the task was not heavily covered during pre-training, leaving sufficient headroom for RL to explore. (ii) the RL data are calibrated to the model's *edge of competence*, neither too easy (in-domain) nor too hard (out-of-domain). When either condition is violated, RL tends to sharpen existing abilities rather than genuinely improve.

**Secondly**, RL incentivizes contextual generalization only when the relevant primitives or skills are present in the base model. Without minimal pre-training exposure to a new context, RL does not induce transfer. But even very sparse coverage (e.g., $\geq$1%) provides a seed that RL can then robustly

reinforce, yielding strong cross-context generalization.

**Thirdly**, introducing a mid-training phase that bridges pre- and post-training distributions *substantially strengthens* both in-domain and out-of-domain performance under a fixed compute budget, highlighting mid-training as an underexplored but powerful lever in training design.

**Fourthly**, process rewards mitigate reward hacking and enhance reasoning fidelity. Incorporating process verification into the reward function aligns reinforcement signals with valid reasoning behavior, leading to measurable improvements in both accuracy and generalization under complex, compositional settings.

## 2. Preliminaries

In this section, we introduce (a) the synthetic *data generation framework* grounded in dependency graphs and contextual rendering that specify the reasoning process, (b) the *task setup* for extrapolative and contextual generalization, and (c) the *process-verified evaluation* framework, which assesses the accuracy of both the reasoning process and the final answer. Together, these components allow us to isolate the distinct effects of pre-training, mid-training, and post-training on reasoning generalization.

### 2.1. Controllable Synthetic Reasoning Dataset

We build on the GSM-Infinite (Zhou et al., 2025b) framework to create a testbed with control over reasoning structure, complexity, and context. Specifically, the data generation pipeline (Figure 2 (a)) involves three key components:

**Dependency Graphs.** Each problem is represented by a direct $\mathcal{G} = (\mathcal{V}, \mathcal{E})$, where nodes $v \in \mathcal{V}$ correspond to variables, and directed edges $e \in \mathcal{E}$ denote dependencies between them. The graph culminates in a designated answer node $v^*$, which yields the final answer $a^*$.

*Figure 2.* **Overview of the data generation framework, task setup, and process-verified evaluation.** The figure depicts the dependency graph $\mathcal{G}$ and contextual templates $\tau$, the task setup for extrapolative and contextual generalization, and the process-verified evaluation framework that checks for correctness of reasoning steps.

**Reasoning Complexity Control.** We quantify the complexity of a graph by the number of arithmetic operations: $\mathrm{op}(\mathcal{G}) = |\mathcal{E}|$, which controls task difficulty from basic arithmetic to complex multi-step reasoning.

**Contextual Rendering.** Given a pre-defined contextual template $\tau$ (e.g., *animals–zoo*, *teachers–school*) with natural language descriptions, we render the dependency graph $\mathcal{G}$ to produce a complete math problem. Finally, we generate diverse math problems by sampling different graphs $\mathcal{G}$ and templates $\tau$, and rendering them into text.

The framework lies in three advantages: 1) *Contamination-free control over training phases.* We specify separate data distributions for pre-, mid-, and post-training to avoid overlap. 2) *Factorized control over structure and context.* Each problem is generated from DAGs, encoding the reasoning structure and dependencies, with numeric values and context instantiated. 3) *Process-level verification.* The ground-truth DAG serves as a reference for verifying intermediate steps and preventing incorrect reasoning. We provide a formulation and explanation in Appendix B.

## 2.2. Task Setup

In real-world deployments, language models usually need to generalize reasoning along two complementary axes: *extrapolative (depth-wise)* and *contextual (breadth-wise)* generalization (Setlur et al., 2025; Zhou et al., 2025a; Huan et al., 2025). Our controlled experiments expose these two dimensions (Figure 2(b)), enabling a precise examination of how *pre-training*, *mid-training*, and *post-training* influence each type of generalization.

**Extrapolative (Depth) Generalization.** This dimension evaluates a model's ability as reasoning depth $\mathrm{op}(\mathcal{G})$ increases (Zhang et al., 2025). A model exhibits strong extrapolative generalization if it can solve problems whose

operation chains exceed those encountered during training.

**Contextual (Breadth) Generalization.** This dimension measures whether a model can transfer its reasoning primitives to novel domains that differ in surface forms but share similar underlying reasoning structure. A model generalizes contextually when its performance remains stable under changes in templates or surface forms while the underlying computation graph remains the same.

Formal notation, dataset construction, and full definitions of the generalization axes are provided in Appendix C.

## 2.3. Evaluation Protocol.

We report all results under a *process-verified evaluation* scheme (Figure 2 (c)). For each instance with ground-truth dependency graph $(\mathcal{G}, a^*)$, the model produces a free-form solution, which we parse into a predicted dependency graph $\hat{\mathcal{G}}$ and final answer $\hat{a}$. The process is evaluated at the step level for each gold node $v \in \mathcal{V}$ by comparing the predicted and ground-truth nodes, their dependencies, and their numeric values. The *process accuracy* is computed as the average step-level accuracy across all gold nodes. A prediction is considered fully correct only when both the reasoning steps and the final answer match. All $pass@k$ metrics (e.g., $pass@1$, $pass@128$) are reported with respect to this strict criterion. Detailed implementation and parsing methods are provided in Appendix D.

## 2.4. Training Setup.

We train 100M-parameter decoder-only Qwen2.5 (Qwen et al., 2025) on a synthetic reasoning corpus generated with GSM-Infinite, which contains 30B tokens spanning multiple operation ranges and contextual templates, and is split into disjoint partitions for pre-training, mid-training, and post-training to avoid contamination across stages.

**Pre-training.** Pre-training provides broad exposure so the model learns general reasoning rules. In our controlled setting, this mainly means acquiring arithmetic primitives and the rules needed to solve GSM-Infinite problems, rather than memorizing open-domain knowledge. Following Chinchilla-style scaling (Hoffmann et al., 2022) and trends in data-rich regimes (Li et al., 2025), we pre-train the 100M model on 10B tokens (100 × parameters). Data covers op=2-10 operations across templates, which is sufficient for the model to master basic reasoning while leaving headroom for harder ranges. The resulting model reaches near-saturated pass@128 accuracy, so later gains on deeper tasks more likely reflect genuine generalization.

**Mid-training.** Mid-training sits after pre-training and has been used to improve later post-training behavior by providing more structured or higher-quality supervision (Liu et al., 2025a; Wang et al., 2025; Akter et al., 2025). It is often implemented with instruction data and next-token or SFT objectives, with the goal of stabilizing optimization and bridging the gap between broad pre-training and reward-driven RL. Here we use a streamlined variant: we keep the same next-token objective as pre-training, but narrow the data distribution toward regimes that resemble RL—where the model shows emerging but incomplete competence. Concentrating supervision at this boundary is intended to strengthen higher-level reasoning priors that RL can then amplify.[2]

**Post-training.** Post-training adapts a pre-trained model to the target task via task-specific objectives. Common approaches are (i) supervised fine-tuning (SFT) on labeled/instruction data and (ii) reinforcement learning (RL) using reward feedback. Because our pre-training corpus is already structured and task-like, we primarily emphasize RL. We apply GRPO (Shao et al., 2024) for post-training and use outcome-rewards that verify the correctness of the response according to the evaluation protocol mentioned in Section 2.3

Appendix F further reports two sets of additional experiments: a 500M Qwen2.5-style scaling study of the controlled setup, and real-world Qwen2.5-7B experiments on math, code, and science benchmarks. These experiments test whether the same conclusions hold for edge-of-competence RL, contextual transfer, mid-/post-training allocation, and process-aware rewards beyond the main 100M setting.

## 3. When Does Post-Training Incentivize Reasoning Beyond the Base Model?

To disentangle the contributions of pre-training and post-training to reasoning capabilities, we isolate the specific impact of RL. We ask: *whether and when RL extends a base*

*model's reasoning capabilities beyond those inherited from pre-training.* By fixing the pre-training stage and varying the difficulty and coverage of post-training data, we identify the specific regimes where RL drives genuine compositional generalization rather than merely amplifying existing skills.

**Task Setting.** We focus on extrapolative generalization (we examine contextual transfer for post-training in Appendix H), defining three problem categories based on operation counts: *In-Distribution (ID)* problems within the pre-training range (op=2-10); *OOD-edge* problems just beyond this range (op=11-14), where the base model retains non-zero pass@128 accuracy; and *OOD-hard* problems substantially beyond the pre-training distribution (op=15-20), where the base model exhibits near-zero accuracy[3]. Solving OOD-hard problems requires composing atomic operations learned from ID data in novel ways to accommodate increased reasoning depth. The experimental setup proceeds as follows:

- **Pre-training:** The base model is pre-trained on 10B tokens consisting of ID problems.

- **Post-training:** We apply GRPO with a total of 200K samples from four distinct difficulty ranges: op=7-10 (ID), op=9-12 (mixed), op=11-14 (edge), and op=17-20 (hard).[4]

---

> **Observation 1**
>
> As shown in Figure 3, the efficacy of post-training is highly sensitive to the pre-training and post-training data regime: (i) For ID tasks (op=2-10), there are obvious performance gains on pass@1 but no improvement on pass@128 regardless of the RL data regime, which indicates that RL only sharpens existing capabilities without extending them. (ii) However, for OOD tasks (op=11-14 and op=15-20), RL always improves pass@128 performance when applied on the *edge of competence* data (op=11-14), demonstrating genuine capability gains beyond pre-training.

---

> **Takeaway 1**
>
> **RL produces true capability gains (pass@128) beyond base models only when two conditions hold:** (i) The task is not heavily covered during pre-training, leaving sufficient headroom for exploration; and (ii) the RL data is calibrated to the model's *edge of competence*, neither too easy ((ID) nor too hard (OOD).

---

[2]Mid-training is only applied in Section 5.

[3]We illustrate this performance ladder in Appendix E.4.

[4]For additional information on the training dynamics and the data recipe, see Appendix G and K.

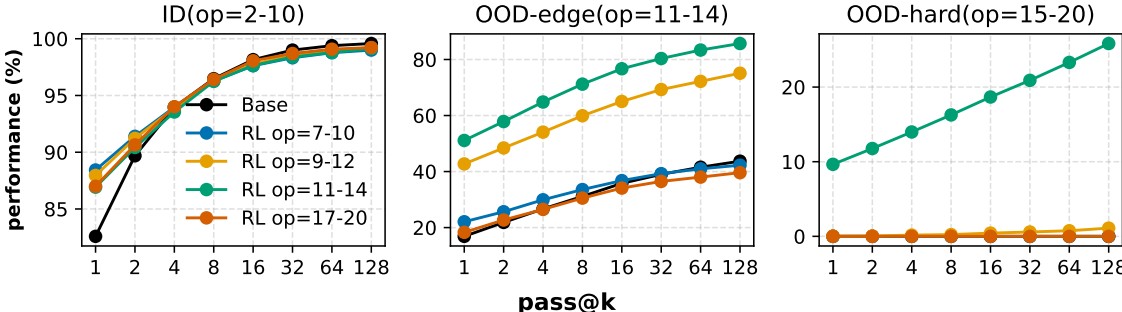

*Figure 3.* `pass@k` performance on three tasks: ID (`op=2-10`), OOD-edge (`op=11-14`), OOD-hard (`op=(15-20)`). RL is applied to four different data regimes (colors). RL on ID tasks never improves beyond the base model at `pass@128`. RL consistently improves `pass@128` on harder tasks when applied beyond the base model's capacity.

---

### Discussion 1

**Connection with recent work.** Recent studies report seemingly conflicting conclusions about whether RL can enhance a base model's reasoning ability. On the one hand, Zhao et al. (2025); Yue et al. (2025) argue that RL does *not* improve `pass@128` accuracy when evaluated on standard tasks such as math and coding—domains that are already well covered during pre-training. On the other hand, work on synthetic tasks with little pre-training coverage (Liu et al., 2025b; Yuan et al., 2025; Sun et al., 2025a) reports substantial post-training gains. Our controlled setting reconciles these findings by showing that they arise from *different regions of the post-training difficulty spectrum*. RL yields no advantage on **in-domain** tasks that the base model already solves, as performance saturates with increasing `pass@k`. In contrast, when RL targets genuinely **OOD** tasks where the base model fails, we observe clear extrapolative improvements, provided the RL data lie near the model's "edge of competence."

### Practical Guidance 1

**Design RL data around the model's *edge of competence*.** We recommend filtering the RL dataset to target tasks where the model fails at `pass@1` but succeeds at `pass@k`. This strategy avoids redundancy on high-`pass@1` tasks while preventing reward sparsity on zero-`pass@k` tasks. This process could also be iterative: we can periodically re-evaluate the pool of "edge of competence" tasks; as the model gets stronger, previously out-of-distribution tasks will drift into the solvability gap, creating a natural, self-paced curriculum.

## 4. How Does Pre-training Exposure Shape Post-Training Generalization?

Having established the conditions under which post-training incentivizes generalization, we turn to a foundational question: *How does pre-training exposure shape post-training*

*generalization?* We hypothesize that pre-training exposure to fundamental reasoning primitives is crucial for effective post-training generalization. To explore this question, with a fixed RL data recipe and setup, we vary the pre-training data to examine its effect on post-training generalization.

**Task Setting.** We focus on contextual generalization to long-tailed *context B* contexts with atomic reasoning primitives (`op=2` examples) during pre-training (experiments on extrapolation are provided in the Appendix I). By manipulating the ratio of *context B* atomic (`op=2`) examples during pre-training, we aim to assess how exposure to these basic primitives shapes the model's ability to transfer learned skills and extrapolate effectively during post-training. Our experimental setup is structured as follows:

- **Pre-training:** The base model is trained on 10B tokens consisting of `op=2-20` *context A* and long-tailed `op=2` *context B* examples. We vary the ratio of atomic `op=2` examples to long-tailed *context B* exposure.

- **Post-training:** RL is applied on 200K samples, consisting of 50% *context A* and 50% *context B*, spanning `op=2-20`. Further details on the training dynamics and data recipe can be found in Appendix J and K.

### Observation 2

As shown in Figure 4, the impact of pre-training exposure to long-tailed contexts on post-training generalization is substantial: (i) When pre-training excludes *context B* or provides no (0%) or very little exposure (0.1%), RL fails to transfer to *context B*. (ii) Introducing even 1% of *context B* data during pre-training significantly enhances post-training generalization even to the hardest tasks of `op=20`. This observation underscores that while RL plays a crucial role in generalization, its effectiveness is heavily dependent on the coverage of the pre-training data, particularly the inclusion of long-tailed contexts.

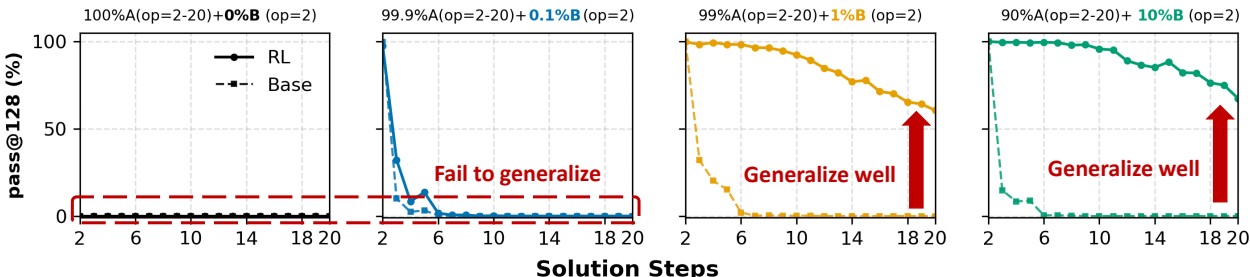

*Figure 4.* `pass@128` performance on *context B* after post-trained with a 50% *context A* + 50% *context B* mixture. Different lines represent levels of pre-training exposure to long-tailed *context B* atomic `op=2` examples. RL incentivizes contextual generalization when the model has minimal exposure (≥1%) to *context B* in pre-training.

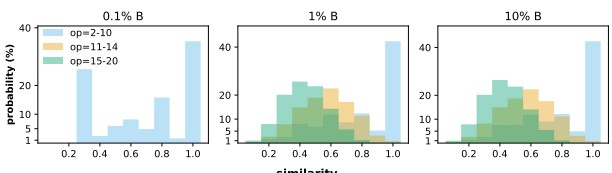

*Figure 5.* Distribution of topological similarity between generated correct *context B* and gold *context A* graphs.

---

### Takeaway 2

**RL incentivizes contextual generalization only when the base model already contains the necessary primitives.** Without minimal pre-training exposure to a new context, RL cannot induce transfer. However, even sparse exposure (e.g., ≥1%) provides a sufficient seed that RL can reinforce during post-training, yielding robust cross-context generalization.

---

### Discussion 2

**Replication or Creation?** Figure 5 illustrates the topological similarity of the generated correct *context B* graphs and the ground-truth from *context A*. High similarity indicates that the model primarily replicates existing *context A* reasoning patterns, while low similarity suggests the emergence of novel reasoning structures distinct from *context A*. We observe effects between task difficulty and exposure: 1) For ID tasks (`op=2-10`), models tend to replicate existing patterns from *context A*. 2) As task complexity increases to OOD (`op=11-20`), models generate more novel structures, especially when pre-trained with sufficient exposure to *context B*.

---

### Practical Guidance 2

**Seed long-tail primitives in pre-training to unlock RL potential.** RL cannot synthesize capabilities from a void; it requires latent "seeds" to amplify. However, these seeds need not be complex. Our results show that RL can extrapolate to hard tasks as long as *atomic reasoning primitives* are present in pre-training. Practitioners should prioritize broad coverage of basic domain

---

knowledge, rules, and skills (at ≈1% density) rather than striving for complex samples. Once these primitives are established, RL acts as a compositor, combining them to solve complex out-of-distribution problems.

## 5. How Does Mid-Training Interact with Post-Training?

While RL effectively enhances extrapolative generalization, its success is often contingent on the representational priors established during pre-training. Recent work (Wang et al., 2025; Liu et al., 2025a) proposes *mid-training* as an intermediate phase between pre-training and post-training, designed to bridge data distributions and strengthen reasoning priors before downstream adaptation.

This raises a key question: *how do mid-training and RL interact under a fixed compute budget, and what balance between them yields the greatest generalization gains?* In this section, we examine the synergy between mid-training and post-training, seeking to define how their interaction drives reasoning generalization.

**Compute Budget Formulation.** For fair comparison, we normalize both phases to *equivalent training tokens* based on flops. For mid-training, the consumption $T_{\text{mid}}$ is the number of supervised tokens processed. For RL, the token-equivalent cost is approximated as:

$$T_{\text{RL}} \approx \tfrac{5}{3} N \cdot r \cdot L_{\text{total}}, \tag{1}$$

where $N$ is the number of RL samples, $r = 6$ the rollout multiplicity, and $L_{\text{total}} = 2048$ the total token length[5].

We systematically vary the RL allocation ratio $\beta \in [0, 1]$ to distribute the **total budget** $T$ between the two phases:

$$T_{\text{mid}} = (1 - \beta) \cdot T, \quad T_{\text{RL}} = \beta \cdot T. \tag{2}$$

**Task Setting.** In this section, we explore the performance of five training configurations using the same base model

---

[5]Detailed budget derivation are provided in Appendix L.1

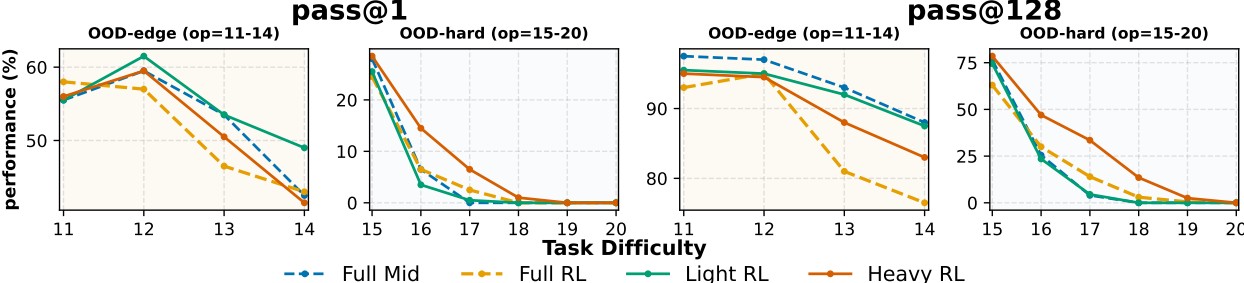

*Figure 6.* `pass@1` and `pass@128` performances on extrapolative tasks under varying mid- and post-training mixture ratios. The data used in mid- and post-training is applied within the OOD-edge ranges. Different lines indicate the compute allocation strategies. Heavy-RL always improves the unseen OOD-hard tasks, while Light-RL improves best `pass@1` on OOD-edge tasks.

pre-trained on 10B `op=2-10` data: **Full mid-training** on 1B supervised tokens from the `op=11-14` range, **Full RL** with 100 steps of batch size 1024 from the same `op=11-14` range, and three mixing strategies—*Light-RL ($\beta = 0.2$)*, *Medium-RL ($\beta = 0.5$)*, and *Heavy-RL ($\beta = 0.8$)*, which balance mid-training and RL under an equivalent compute budget. The compute budget formulation in Section 5 allows for a direct comparison of data mixture strategies. Detailed training setup can be found in Appendix L.

---

### Observation 3

As shown in Figure 6, compute allocation induces qualitatively different behaviors across the generalization spectrum. (1) On **OOD-edge** tasks, configurations with *full mid-training* and *light RL* outperform those with *heavy* or *full* RL, with light RL achieving the best `pass@1` performance. (2) For **OOD-hard** tasks, reallocating more budget toward heavy RL substantially improves performance on the hardest instances in both `pass@1` and `pass@128`. These trends suggest that RL-driven exploration is indispensable for generalizing to harder tasks, but a substantial mid-training allocation remains critical for instilling the priors that RL can effectively exploit. We further analyze the impact of varying compute budgets in Appendix L.

---

### Takeaway 3

**Introducing a mid-training phase that bridges pre- and post-training distributions substantially strengthens generalization under a fixed compute budget.** This highlights mid-training as an underexplored but powerful lever in training design. Compute should be allocated in a task-aware manner: (i) when prioritizing in-distribution performance, allocate more budget to mid-training with only light RL; (ii) for out-of-distribution generalization, reserve a modest portion of compute for mid-training to establish essential priors, and dedicate the remaining budget to heavier RL exploration.

---

### Discussion 3

**The Role of Mid-Training.** Recent work (Shao et al., 2025; Gandhi et al., 2025) finds that some bases (e.g., Qwen (Qwen et al., 2025)) respond much better to RL than others (e.g., LLaMA (Touvron et al., 2023)). A converging explanation is *mid-training*: an intermediate stage that brings supervision closer to the post-training/RL distribution. Reasoning-focused mid-training substantially improves RL readiness: Wang et al. (2025) show that LLaMA models mid-trained on structured reasoning can match the RL performance of stronger Qwen bases. More broadly, Liu et al. (2025a) argue mid-training bridges the pretraining–posttraining gap, reducing forgetting and easing adaptation, consistent with the "frontloading" view that earlier structured supervision provides scaffolding that RL can efficiently amplify (Akter et al., 2025). Overall, mid-training appears central for stable, sample-efficient RL gains beyond merely sharpening existing abilities.

---

### Practical Guidance 3

**Balance mid-training and post-training around complementary strengths.** Design the training pipeline by treating mid-training as the phase for *installing priors* and RL as the phase for *scaling exploration*. For mid-training, curate datasets that lie at the model's "edge of competence", which stabilizes the primitives required for RL. Practitioners should adjust the compute budget based on the deployment goal: (1) For **reliability** on similar tasks (OOD-edge), allocate the majority of compute to mid-training and use light RL. (2) For **exploration** on complex tasks (OOD-hard), allocate mid-training to a modest budget (sufficient only to establish priors) and spend heavy compute on RL exploration.

## 6. Mitigating Reward Hacking via Process Supervision in Outcome Rewards

Post-training with outcome-based rewards has proven highly effective in improving reasoning performance, yet it remains

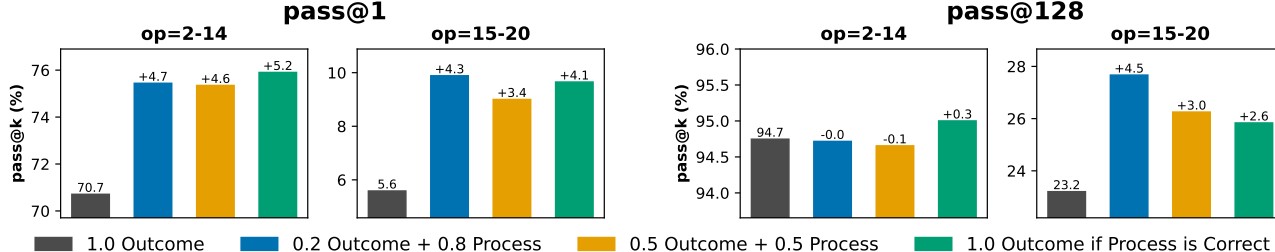

*Figure 7.* `pass@k` performance under different rewards. Each bar corresponds to a distinct reward-mixing strategy. Incorporating process-level information into the outcome reward consistently yields measurable performance gains across evaluation settings.

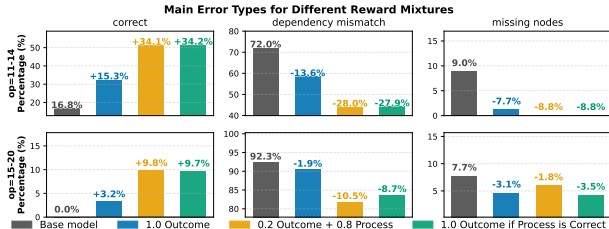

*Figure 8.* **Effects of reward mixtures on reasoning correctness and structural error types.** Process-aware reward mixtures improve correctness while reducing dominant structural failures, especially dependency mismatches, under OOD-edge and OOD-hard evaluation.

vulnerable to *reward hacking*—a failure mode where models achieve high final accuracy by exploiting spurious shortcuts or producing correct answers through invalid reasoning chains. Earlier, we introduced *process verification* as an evaluation criterion that rewards models only when both intermediate steps and the final outcome are correct. Here, we extend this principle into the *reward design* itself, asking: *Can process-aware supervision mitigate reward hacking while preserving generalization performance?*

**Task Setting.** Under the setting in Section 3, we post-train with `op=11-14` data recipe and evaluate on `op=2-14` and `op=15-20` with two main reward mixtures:

Directly integrating process verification into outcome reward function:

$$R = \alpha R_{\text{out}} + (1 - \alpha) R_{\text{pv}}. \quad (3)$$

$R_{\text{out}}$ denotes the traditional outcome-based reward (1 for a correct final answer, 0 otherwise), which may be sparse and susceptible to outcome reward hacking. $R_{\text{pv}}$ represents the process verification reward defined by the process-level accuracy criteria in Appendix D, which is a dense reward reflecting the reasoning step correctness. $\alpha \in [0, 1]$ controls the balance between outcome accuracy and process fidelity.

Another formulation we consider is a stricter and sparse formulation:

$$R = \begin{cases} R_{\text{out}}, & \text{if } R_{\text{pv}} = 1, \\ 0, & \text{otherwise,} \end{cases}$$

which grants outcome rewards only when the entire reasoning is verified as correct. This setup does not change the sparse formulation but provide more strict supervisions.

**Observation 4**

As shown in Figure 7, integrating process verification notably improves `pass@1` by 4–5% across extrapolative (`op=15-20`) settings. Moderate reward mixes $(0.2, R_{\text{out}} + 0.8, R_{\text{pv}})$ achieve the best balance between outcome accuracy and reasoning consistency, while the strict reward ($R_{\text{out}}$ only if $R_{\text{pv}}=1$) further enhances substantial improvements. These results confirm that process-level supervision effectively mitigates reward hacking and encourages faithful reasoning behavior.

**Takeaway 4**

**Process-aware rewards mitigate reward hacking and enhance reasoning fidelity.** Incorporating process verification into the reward function aligns reinforcement signals with valid reasoning behavior, leading to measurable improvements in both accuracy and generalization under complex, compositional settings.

**Discussion 4**

**How does process verification reshape RL generalization?** Process-aware rewards materially change *how* the policy behaves: they reduce structural failure modes (especially dependency mismatches), indicating a shift from shortcut exploitation toward faithful intermediate reasoning. In our reward-mixture ablations, adding dense process supervision yields the largest drop in the dominant structural error type and the best accuracy, with the advantage most pronounced under extrapolation `op=15-20` (see Figure 8). Therefore, aligning RL rewards with valid reasoning traces is important for improving OOD generalization.

**Practical Guidance 4**

**Combine sparse outcome signals with dense process-level feedback.** In practice, blending sparse final-outcome signal with richer process-level information is

beneficial (Gunjal et al., 2025; Khalifa et al., 2025). Provided that the process supervision is of high quality (Cui et al., 2025), we recommend incorporating process-level information into the outcome reward. This helps mitigate reward-hacking and consistently improves performance.

## 7. Conclusion

In this work, we presented a controlled investigation into how pre-training and post-training jointly determine the reasoning capabilities of language models. By disentangling the contributions of each stage, our study clarifies the causal mechanisms through which RL enhances or fails to enhance reasoning generalization. Using fully controllable synthetic reasoning tasks and process-level evaluations, we demonstrated that genuine reasoning improvements through post-training arise only when key reasoning primitives are established during pre-training. Together with experiments on 500M scaling and Qwen2.5-7B real-world benchmarks, these results refine our understanding of reasoning development in language models and provide actionable guidance for constructing data curricula, designing reward functions, and allocating compute across training stages.

## Impact Statement

This paper studies how pre-training, mid-training, and RL interact in reasoning language models. By using controlled experiments, we aim to provide empirical guidance for designing more efficient, interpretable, and predictable training pipelines, rather than introducing a new capability-amplifying method.

Our findings also have safety implications. RL can improve reasoning, but misspecified rewards may encourage reward hacking or unfaithful reasoning. Process-level rewards can mitigate some failures, but their reliability depends on verifier quality and coverage. We therefore view this work as supporting more careful evaluation, robust oversight, and alignment-aware RL objectives.

## Acknowledgment

The authors would like to thank Kai Zhang, Yuetai Li, Ge Zhang, Boshi Wang, Seungone Kim, Yuanzhi Li, Xinyu Yang, Yao Fu, Ziqiao Ma, Jinjie Ni, and Junyang Lin for their constructive feedback and comments on the early draft of the paper. Xiang Yue was supported in part by a Carnegie Bosch Institute Fellowship.

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

# A. Related Work

**RL Generalization of Reasoning LMs.** The role of RL in driving generalization in LMs has been the subject of extensive discussion. Recent work presents differing views on whether RL can extend reasoning beyond the capabilities of the base model, with contrasting arguments emerging in the literature.

On the one hand, several studies caution against overestimating RL's ability to push the boundaries of a base model. Yue et al. (2025) argue that while RL-trained models may outperform base models at small values of pass@k (e.g., k = 1), the performance advantage diminishes as k increases (e.g., k = 128). Their coverage and perplexity analyses suggest that the reasoning capabilities of RL-trained models remain ultimately constrained by the base model's representational capacity. Additionally, Wu et al. (2025) provides a theoretical framework asserting that RL cannot surpass the base model's inherent limitations, thus challenging the notion that RL can enable new, generalizable reasoning skills.

On the other hand, there are strong arguments in favor of RL's ability to enable generalization, particularly in tasks where the base model performs poorly. Liu et al. (2025b) highlights the success of ProRL in improving performance on synthesized reasoning tasks, where base models demonstrate significant limitations. Further supporting this view, Sun et al. (2025a;b) provides clear evidence of RL's potential to induce novel strategies for complex problem families. Yuan et al. (2025) propose a synthetic function composition task, demonstrating that RL-trained models can generalize to unseen function compositions that the base model cannot handle.

In our work, we contribute to this ongoing debate by providing empirical evidence that the two perspectives are not mutually exclusive. Instead, we show that the conditions under which RL can drive generalization are nuanced and depend on the base model's reasoning primitives as well as the nature of the post-training data used during RL fine-tuning.

**Understanding LMs via Controlled Experiments.** Several prior work Yuan et al. (2025); Liu et al. (2025b); Sun et al. (2025a) has emphasized the importance of controlled experiments in understanding the capabilities of LMs. However, this line of work mainly focuses on synthetic tasks designed for post-training RL, which may not fully capture the complexities of the full spectrum of reasoning tasks from pre-training to post-training. Especially in the context of reasoning tasks, controlled settings allow researchers to isolate specific factors, e.g., data contamination, random-guess answers, as well as controlling the reasoning primitives for different training phases. We build upon this line of work by designing controlled experiments motivated by Ye et al. (2024) to synthesize GSM-style reasoning tasks (Cobbe et al., 2021; Liu et al., 2023; Mirzadeh et al., 2025; Zhou et al., 2025b)

# B. Data Generation Framework

This section provides the formal details of the controllable data generation framework used throughout the paper. We describe (i) the graph-level formalism underlying each reasoning instance, (ii) the abstraction mechanism that separates structure from numeric and linguistic instantiations, (iii) the contextual rendering function that maps graphs to natural-language problems, and (iv) the concrete generation pipeline and deduplication procedure.

## B.1. Graph-Level Formalism

Each reasoning instance is grounded in a directed acyclic graph (DAG)

$$\mathcal{G} = (\mathcal{V}, \mathcal{E}),$$

where each node $v_i \in \mathcal{V}$ represents a latent quantity (e.g., "number of adult lions") and each directed edge $(v_j \rightarrow v_i) \in \mathcal{E}$ encodes a functional dependency. We restrict dependencies to elementary arithmetic operations:

$$v_i = f_i\big((v_j)_{j \in \mathrm{pa}(i)}\big), \qquad f_i \in \{+, -, \times, \div\},$$

where $\mathrm{pa}(i)$ is the parent set of node $i$.

Given numeric assignments to all leaf nodes, we define an evaluation map

$$\mathrm{val} : \mathcal{V} \rightarrow \mathbb{R}$$

recursively by

$$\mathrm{val}(v_i) = f_i\big(\{\mathrm{val}(v_j)\}_{j \in \mathrm{pa}(i)}\big),$$

with base cases given by the leaf values. For a designated query node $v^*$, the ground-truth answer is

$$a^* := \mathrm{val}(v^*).$$

In the GSM-Infinite implementation that we build upon (Zhou et al., 2025b), the query node $v^*$ corresponds to:

- the last numeric node in the topological order of the *forward* generator, or

- the distinguished unknown parameter in the *equation-style reverse* generator.

Throughout, the DAG $\mathcal{G}$ is treated as the symbolic reasoning graph whose structure is shared across different numerical instantiations and linguistic realizations.

**Reasoning Complexity.** We quantify the structural complexity of an instance by the number of arithmetic operations:

$$\mathrm{op}(\mathcal{G}) = |\mathcal{E}|.$$

This quantity lower-bounds the minimal length of the compositional reasoning chain needed to compute $a^*$, and is the primary knob we vary when studying extrapolative (depth-wise) generalization.

### B.2. Abstract and Instance Parameters

Following the abstraction mechanism of GSM-Infinite, we explicitly separate *structure*, *numeric instantiation*, and *linguistic context*.

**Abstract Parameters.** Each graph $\mathcal{G}$ is associated with a set of *abstract parameters* that:

- specify which variables exist and how they decompose (e.g., that "total animals" decomposes into "lions" and "elephants"), and

- determine the edge set $\mathcal{E}$ and the operation $f_i$ attached to each node.

These parameters define a purely symbolic graph, independent of particular numbers or entities.

**Instance Parameters.** Given an abstract graph, *instance parameters* instantiate it with concrete values and entities:

- numeric assignments to leaf nodes (e.g., "there are 12 adult lions and 7 elephant calves"), and

- bindings of variables to context-specific surface forms (e.g., "adult lions in the city zoo").

Instantiating different numeric values on the same abstract graph leads to a family of structurally identical problems that differ only in their concrete numbers.

**Implicit Reasoning.** Not all abstract dependencies need to be explicitly verbalized in the natural-language problem. For a given linguistic rendering, the edge set can be partitioned as

$$\mathcal{E} = \mathcal{E}_{\text{explicit}} \cup \mathcal{E}_{\text{implicit}}, \qquad \mathcal{E}_{\text{explicit}} \cap \mathcal{E}_{\text{implicit}} = \emptyset,$$

where $(v_j \to v_i) \in \mathcal{E}_{\text{explicit}}$ denotes a relation that is directly stated in the text (e.g., "there are 5 more elephants than lions"), while $(v_j \to v_i) \in \mathcal{E}_{\text{implicit}}$ denotes a relation that is part of the ground-truth reasoning graph but never directly verbalized (e.g., "total animals equals lions plus elephants"). This separation allows explicit and implicit reasoning steps to coexist within the same underlying graph and enables us to probe models' ability to recover unspoken dependencies.

### B.3. Contextual Rendering

To map a symbolic graph to a natural-language problem, we introduce a contextual rendering function

$$\Phi : (\mathcal{G}, \tau) \mapsto x,$$

where $\tau \in \mathcal{T}$ is a *contextual template* and $x$ is the resulting text instance.

**Templates.** A template $\tau$ (e.g., *animals–zoo*, *teachers–school*, *movie-festival*) specifies:

- how abstract variables are lexicalized into domain-specific surface forms (e.g., "adult lions", "children in class A", "tickets sold on day 1"), and

- which subset of edges is realized explicitly in the wording, thereby determining the split between $\mathcal{E}_{\text{explicit}}$ and $\mathcal{E}_{\text{implicit}}$.

For any two templates $\tau_a, \tau_b \in \mathcal{T}$ that differ only in surface context, the induced problems remain structurally identical:

$$\text{Struct}(\Phi(\mathcal{G}, \tau_a)) = \text{Struct}(\Phi(\mathcal{G}, \tau_b)), \quad \forall \tau_a, \tau_b \in \mathcal{T},$$

even though their surface realizations, entities, and explicit/implicit splits may differ. Thus, a single abstract graph can be rendered into semantically distinct yet structurally equivalent problems, which we leverage to study contextual (breadth-wise) generalization.

**Solution Format.** The rendering function produces a triple

$$x = \big([\text{question}], [\text{solution}], [\text{answer}]\big),$$

where:

- **[question]** is the natural-language representation of the problem posed by the symbolic graph $\mathcal{G}$, typically including a query regarding some aspect of the graph (e.g., "How many tickets were sold on day 1?"). It abstracts away the underlying structure and provides the context for the solution.

- **[solution]** is a step-by-step derivation that follows the topological order of the symbolic graph $\mathcal{G}$. It includes intermediate reasoning steps and logical connections between the graph's elements, ultimately leading to the final answer. The solution explicitly shows how each part of the problem is derived or calculated.

- **[answer]** is the final response to the query posed in the [question], derived through the [solution] process. It is typically a numerical value or a specific entity that answers the question posed.

This structure ensures that the rendered output is both human-readable and logically consistent with the underlying symbolic graph, maintaining the integrity of the original problem while making it accessible in natural language.

**Example Rendered Instance for *teachers–school* Context**

**[Question]**

**[question]**
The number of elementary school in Westhaven City equals the number of public highschool in Westhaven City. The number of elementary school in Evervale City equals the sum of the number of public highschool in Evervale City and the number of regional medical school in Westhaven City. The total number of schools in Evervale City equals 22. The number of elementary school in Brightford equals 3. The number of public highschool in Brightford equals 2. The number of regional medical school in Brightford equals the total number of schools in Westhaven City. The number of regional medical school in Westhaven City equals 2. The number of regional medical school in Evervale City equals 2 times the number of regional medical school in Brightford. The number of public highschool in Westhaven City equals 3. The number of public highschool in Evervale City exists, and its number is greater than 0.
How many public highschool does Evervale City have?
**[/question]**

**Solution**

**[solution]**
The question is difficult, so we use equations to solve it.
Define public highschool in Westhaven City as $U$; so $U = 3$. Define elementary school in Westhaven City as $B$; so $B = U = 3$. Define regional medical school in Westhaven City as $h$; so $h = 2$.
Define total number of schools in Westhaven City as $y$; $d = U + B = 3 + 3 = 6$, so $y = d + h = 6 + 2 = 8$.
Define regional medical school in Brightford as $Q$; so $Q = y = 8$. Define regional medical school in Evervale City as $S$; $z = Q = 8$, so $S = 2z = 16$.
Define public highschool in Evervale City as $x$ (unknown). Define elementary school in Evervale City as $m$; so $m = x + h = x + 2$. Define total number of schools in Evervale City as $k$.

$$n = x + (x + 2) = 2x + 2, \qquad k = n + S = 2x + 18.$$

Since $k = 22$:

$$2x + 18 = 22, \quad 2x = 4, \quad x = 2.$$

**[/solution]**

**Answer**

**[answer]** 2 **[/answer]**

### B.4. Generation Pipeline and Structural Knobs

Our data generator follows a stage-wise procedure reminiscent of GSM-Infinite forward and reverse generators:

1. **Structural sampling.** We first sample structural knobs that define the dependency graph:

   - a target operation count range for $op(\mathcal{G})$;
   - graph shape parameters (e.g., allowable in-degree, layering pattern) that control fan-in and depth; and
   - operation types $f_i \in \{+, -, \times, \div\}$ attached to nodes.

   These choices determine a layered DAG $\mathcal{G}$ with a unique query node $v^*$.

2. **Abstract and instance parameterization.** Given $\mathcal{G}$, we sample abstract parameters (variable roles and decompositions) and instance parameters (numeric values on leaves) and evaluate all node values in topological order using the evaluation map $\mathrm{val}$ defined above.

3. **Contextual rendering.** We choose a template $\tau \in \mathcal{T}$ and apply the rendering function $\Phi(\mathcal{G}, \tau)$ to obtain a natural-language triple (problem, question, solution), deciding which dependencies are verbalized (explicit) and which remain implicit.

4. **Forward vs. reverse modes.** Following (Zhou et al., 2025b), we support two modes of generation: In *forward* mode, we generate a standard arithmetic word problem where the final node in the topological order is queried. In *reverse* mode, we treat one node as an unknown and phrase an equation-style problem where the model must solve for that quantity, while the rest of the graph remains fully specified.

By jointly varying (i) the operation count $op(\mathcal{G})$ and (ii) the template $\tau$, we obtain a clean two-dimensional testbed for studying depth scaling and context transfer. The same framework is used to define distinct data distributions for pre-training, mid-training, and post-training by sampling from different regions of $(op(\mathcal{G}), \tau)$-space.

### B.5. Deduplication and Canonicalization

To guarantee cleanliness and avoid contamination across training and evaluation splits, we perform exact hash-based deduplication at the level of rendered triples. Each instance is canonicalized by:

- serializing the triple (problem, question, solution) into a normalized string representation (e.g., stripping extraneous whitespace and normalizing numeric formatting), and

- hashing this canonical form to obtain a global identifier.

We discard any duplicate hashes within and across splits, ensuring that no identical problem–solution triple appears in both training and evaluation.

# C. Task Setup

In real-world deployments, language models are expected to generalize reasoning along two complementary dimensions (Setlur et al., 2025; Zhou et al., 2025a; Huan et al., 2025). Our controllable dataset makes these dimensions explicit and allows us to probe how *pre-training*, *mid-training*, and *post-training* shape each type of generalization.

**Notation.** Let $f_\theta^{\text{pre}}$, $f_\theta^{\text{mid}}$, and $f_\theta^{\text{post}}$ denote the language models after pre-training, after additional mid-training, and after post-training (RL), respectively. We write $\text{Correct}(f, \mathcal{G}, \tau)$ for correctness on instances generated from graph $\mathcal{G}$ under template $\tau$, using the strict metric defined in the evaluation protocol below.

**Extrapolative (Depth) Generalization.** We parameterize each training phase $\phi \in \{\text{pre}, \text{mid}, \text{post}\}$ by the range of operation counts it sees. Let $\mathcal{O}_\phi$ be the set of $\text{op}(\mathcal{G})$ values present in the training distribution of phase $\phi$, and let

$$\mathcal{O}_{\text{train}} = \mathcal{O}_{\text{pre}} \cup \mathcal{O}_{\text{mid}} \cup \mathcal{O}_{\text{post}}.$$

An *in-distribution* evaluation condition uses graphs with $\text{op}(\mathcal{G}) \in \mathcal{O}_{\text{train}}$, while an *extrapolative* (out-of-distribution, OOD) condition evaluates on graphs with

$$\text{op}(\mathcal{G}) > \max \mathcal{O}_{\text{train}}.$$

A model exhibits extrapolative generalization if it maintains high process-verified accuracy on these longer, unseen operations while remaining stable on in-distribution ones. By the varied difficulty ranges populate $\mathcal{O}_{\text{pre}}$, $\mathcal{O}_{\text{mid}}$, and $\mathcal{O}_{\text{post}}$, we can isolate how each phase contributes to depth-wise generalization.

**Contextual (Breadth) Generalization.** A fixed reasoning graph $\mathcal{G}$ can be rendered into structurally equivalent instances under different templates,

$$\text{Struct}(\Phi(\mathcal{G}, \tau_a)) = \text{Struct}(\Phi(\mathcal{G}, \tau_b)) \quad \text{in principle,}$$

Our dataset is *randomly sampled* during training and does not deliberately align graphs across templates. As a result, most graphs are observed only under a subset of contexts during training. Let $\mathcal{T}_\phi^{\text{train}}$ denote the templates exposed during training phase $\phi$, and $\mathcal{T}^{\text{eval}}$ the broader evaluation pool, including long-tailed templates. A model at phase $\phi$ demonstrates contextual generalization if it preserves reasoning performance when the narrative surface form shifts, even when the new context was never encountered during training:

$$\text{Acc}(f_\theta^\phi, \mathcal{G}, \tau_a) \approx \text{Acc}(f_\theta^\phi, \mathcal{G}, \tau_b), \qquad \tau_b \notin \mathcal{T}_\phi^{\text{train}}.$$

Under this setup, contextual generalization measures whether the model has learned transferable *reasoning primitives* rather than memorized task styles, allowing it to apply the same structural reasoning across known, unseen, and long-tailed narrative environments.

## D. Process-Verified Evaluation

Given an input instance with ground-truth graph $(\mathcal{G}, a^*)$, the model produces a free-form solution $s$. We deterministically parse $s$ into a predicted dependency graph

$$\hat{\mathcal{G}} = (\hat{\mathcal{V}}, \hat{\mathcal{E}}, \widehat{\text{val}}), \qquad \hat{a},$$

where nodes in $\hat{\mathcal{V}}$ correspond to named intermediate quantities in the solution, $\hat{\mathcal{E}}$ encodes which previously defined quantities each step depends on, $\widehat{\text{val}}$ stores the inferred numeric value for each node, and $\hat{a}$ is the extracted final answer. The parser segments the solution into "Define . . . as . . . " steps, infers each step's dependencies from the variables it uses, and evaluates the last computable arithmetic expression in the step (falling back to the last numeric literal if needed) to obtain a numeric value. This yields a graph-level representation of the model's reasoning trace aligned with the gold dependency graph.

Let the gold graph be

$$\mathcal{G} = (\mathcal{V}, \mathcal{E}, \text{val}), \qquad a^*,$$

with node set $\mathcal{V}$, edge set $\mathcal{E}$, and value map val. We evaluate the reasoning process at the *step level*. For each gold node $v \in \mathcal{V}$, define a per-step correctness indicator

$$s(v; \hat{\mathcal{G}}, \mathcal{G}) = \begin{cases} 1, & \text{if } v \in \hat{\mathcal{V}}, \ \text{pa}_{\hat{\mathcal{G}}}(v) = \text{pa}_{\mathcal{G}}(v), \text{ and} \\ & \quad \text{val}(v), \widehat{\text{val}}(v) \text{ are both defined and } \widehat{\text{val}}(v) = \text{val}(v), \\ 0, & \text{otherwise,} \end{cases}$$

where $\text{pa}_{\mathcal{G}}(v)$ and $\text{pa}_{\hat{\mathcal{G}}}(v)$ denote the parent sets (dependencies) of $v$ in the gold and predicted graphs, respectively. Missing nodes, incorrect dependency sets, or mismatched values all yield $s(v; \hat{\mathcal{G}}, \mathcal{G}) = 0$.

We then define the *process accuracy* of a predicted reasoning trace as the average step-level accuracy over all gold nodes:

$$\text{ProcessAcc}(\hat{\mathcal{G}}; \mathcal{G}) = \frac{1}{|\mathcal{V}|} \sum_{v \in \mathcal{V}} s(v; \hat{\mathcal{G}}, \mathcal{G}).$$

Extra predicted nodes $v \in \hat{\mathcal{V}} \setminus \mathcal{V}$ are allowed and do not affect ProcessAcc; they correspond to redundant but compatible intermediate steps.

A prediction is regarded as fully correct only when both the reasoning graph and the final answer match. We formalize this via a *verified correctness*:

$$\text{VerifiedCorrect}(\hat{a}, \hat{\mathcal{G}}; a^*, \mathcal{G}) = \begin{cases} 1, & \text{if ProcessAcc}(\hat{\mathcal{G}}; \mathcal{G}) = 1 \text{ and } \hat{a} = a^*, \\ 0, & \text{otherwise.} \end{cases}$$

Accordingly, all `pass@k` metrics (e.g., `pass@1`, `pass@128`) reported in this work treat a sample as correct only when the model (i) predicts every gold step correctly (step-level process accuracy = 1) and (ii) produces the correct final answer. This strict criterion ensures that reported gains reflect genuine, faithful reasoning rather than coincidental correctness.

# E. Training Setup

## E.1. Model Architecture

We conduct experiments using decoder-only Qwen2.5 Architecture (Qwen et al., 2025) models with 100M parameters. The detailed architecture configurations are as in Table 1

| Component | Configuration |
|---|---|
| Model Type | Qwen2.5 |
| Number of Layers | 12 |
| Hidden Size | 768 |
| Intermediate Size | 3,072 |
| Number of Attention Heads | 12 |
| Number of Key-Value Heads | 2 |
| Activation Function | SiLU |
| RMS Norm Epsilon | 1e-06 |

*Table 1.* Model architecture details for the 100M-parameter Qwen2.5 model used in experiments.

## E.2. Tokenizer and Input Representation

We follow the *Physics of Language Models* series (Ye et al., 2024) and train a byte-pair encoding (BPE) tokenizer directly on our synthetic reasoning corpus. The resulting vocabulary has 2,200 tokens (including special tokens). All problems, questions, and solutions are tokenized with a maximum sequence length of 2,048 tokens.

## E.3. Hyperparameters

**Pre-training.** All experiments start from a 100M-parameter Qwen2.5 model trained from scratch on our controllable reasoning corpus, using a $100\times$ token-to-parameter ratio, pre-training on 10B tokens. We use a context length of 2048 tokens, batch-size 512K tokens, learning rate $2 \times 10^{-4}$ with weight decay 0.1, cosine decay with minimum learning rate $3 \times 10^{-5}$, warmup ratio 5%, and a single epoch over the corpus. All models are trained in bf16 precision.

**Mid-training (Continue Pre-training).** Starting from the pre-trained checkpoint, we perform an additional and optional curriculum in § 5. We train with maximum sequence length 2,048. We use a global batch size of 512K tokens, learning rate $1 \times 10^{-4}$, weight decay 0.1, cosine decay with minimum learning rate $3 \times 10^{-5}$, and a higher warmup ratio of 15%.

**Post-training.** Finally, we apply RL fine-tuning usin GRPO (Shao et al., 2024). We use a global batch size of 1,024 examples, maximum prompt and response lengths of 1024 tokens, and two training epochs. The actor uses learning rate $1 \times 10^{-6}$, PPO mini-batch size 256, micro-batch size 16 per GPU, KL regularization with coefficient $10^{-3}$ (low-variance KL penalty), and zero entropy bonus. During RL rollouts we sample with temperature $T_{\text{RL}} = 1.0$, top-$p = 1.0$, and no top-$k$ truncation (full nucleus sampling). For offline evaluation and reporting we generate with temperature $T_{\text{eval}} = 0.7$, top-$p = 1.0$, and top-$k = -1$ (no truncation), using a maximum of 1,024 new tokens per problem.

## E.4. Performance Ladder

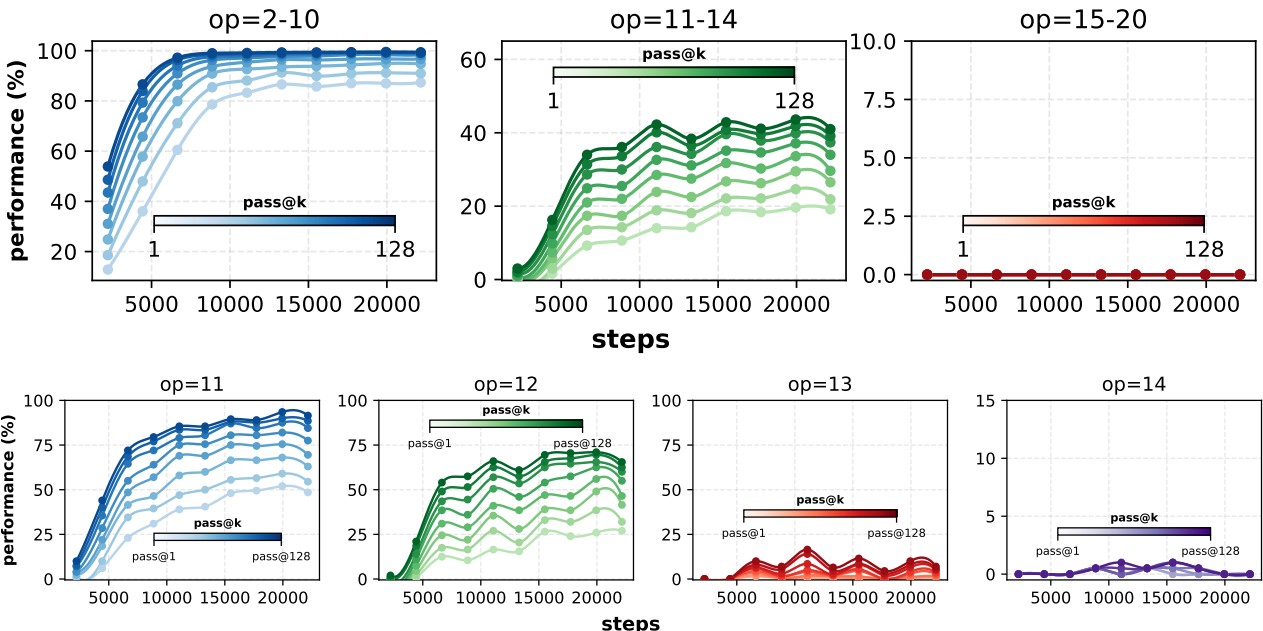

*Figure 9.* Pre-training dynamics across varying operation ranges: In-distribution tasks (op=2-10), edge-of-competence OOD tasks (op=11-14), and OOD-hard tasks (op=15-20). The plots show the performance measured by pass@k over training steps.

The performance ladder defines three key levels based on task difficulty: 1)**In-distribution tasks** (op=2-10): Aim for near-100% pass@128 accuracy; 2)**OOD-edge tasks** (op=11-14): Ensure non-zero pass@128 performance; 3) **OOD-hards tasks** (op=15-20): Aim for zero pass@128, signaling the model's competence limits. Post-training is performed on the edge of competence, ensuring the model generalizes to harder tasks. A breakdown of training dynamics across these performance levels is shown in Figure 9.

*Table 2.* Scaling the controlled edge-of-competence experiment to a 500M Qwen2.5-style model. RL on near-edge data gives the strongest extrapolative improvement on `op=15-20`, while RL on already-easy or very-hard data is less effective. Parentheses report absolute change over Base for each difficulty range.

| Setting | op=2-10 | | op=11-14 | | op=15-20 | |
|---|---|---|---|---|---|---|
| | pass@1 | pass@128 | pass@1 | pass@128 | pass@1 | pass@128 |
| Base | 96.6 | 99.8 | 47.4 | 77.9 | 14.6 | 45.3 |
| RL op=7-10 | 90.6 (-6.0) | 99.5 (-0.3) | 40.3 (-7.1) | 75.0 (-2.9) | 16.0 (+1.4) | 48.9 (+3.6) |
| RL op=9-12 | 90.4 (-6.2) | 99.6 (-0.2) | 53.0 (+5.6) | **88.6 (+10.7)** | 23.7 (+9.1) | 49.6 (+4.3) |
| RL op=11-14 | 90.3 (-6.3) | 99.5 (-0.3) | **53.4 (+6.0)** | **88.6 (+10.7)** | **26.2 (+11.6)** | **53.2 (+7.9)** |
| RL op=17-20 | 90.3 (-6.3) | 99.6 (-0.2) | 40.6 (-6.8) | 76.3 (-1.6) | 16.3 (+1.7) | 49.3 (+4.0) |

# F. Additional Experiments

To further test the robustness of our findings beyond the main 100M controlled setting, we conduct two groups of additional experiments. First, we scale the controlled synthetic setup to a 500M Qwen2.5-style decoder-only model while keeping the same task construction, operation-count difficulty control, and evaluation protocol. In the controlled experiments, `op=2-10` denotes easier in-distribution tasks, `op=11-14` denotes edge-of-competence tasks, and `op=15-20` denotes harder extrapolative tasks. Second, we evaluate whether the same qualitative trends extend to more naturalistic reasoning settings using Qwen2.5-7B Base on math, code, and science benchmarks. Unless otherwise specified, we report `pass@k`, where higher values indicate better performance.

## F.1. Scaling the Controlled Setup to 500M

**Setup.**    We first repeat our controlled experiments with a 500M Qwen2.5-style model. The model is trained and evaluated under the same synthetic reasoning setup as in the main paper. We use operation count as the controlled difficulty variable and evaluate performance on three difficulty ranges: `op=2-10`, `op=11-14`, and `op=15-20`. We use BASE to denote the pre-trained model before RL. An entry such as "RL `op=11-14`" means that the RL data are sampled from tasks with 11–14 atomic operations.

**RL is most effective near the edge of competence.**    Table 2 shows that the main conclusion of our 100M experiments remains consistent at 500M scale. RL on tasks that are already easy mainly preserves high in-distribution `pass@128` while reducing `pass@1`, suggesting that it changes the sampling distribution more than the underlying coverage. In contrast, RL on tasks near the model's edge of competence yields the strongest extrapolative improvements. For example, on `op=15-20`, RL on `op=11-14` improves over the base model from 14.6 to 26.2 `pass@1` and from 45.3 to 53.2 `pass@128`. RL on easier or much harder data is less effective, suggesting that the RL signal is most useful when the training data sit near the model's current reach. This directly mirrors the main-text conclusion in Section 3: RL produces genuine extrapolative gains only when the evaluation task has headroom and the RL data are calibrated to the model's edge of competence.

**Contextual transfer still requires non-zero exposure.**    We next test whether contextual transfer remains dependent on minimal pre-training exposure. Here, *context B* denotes a target semantic context that shares the same underlying reasoning structure as the source context but differs in surface semantics. We vary the amount of *context B* exposure during pre-training and evaluate on *context B* after RL. Table 3 shows that contextual transfer still requires minimal pre-training exposure: with 0% exposure to *context B*, RL fails completely, while even sparse exposure (1–10%) already enables strong transfer. The 0.1% setting gives a weaker but non-zero signal, further supporting the conclusion that RL needs at least some pre-training grounding in the target context. This links back to Section 4, where the main text shows that RL can transfer to a new context only when pre-training has seeded the relevant primitives.

**Mid-training and RL remain complementary.**    We further examine compute allocation between mid-training and RL at 500M scale. We fix the total training budget and vary the fraction allocated to RL. For example, "60% RL" means that 60% of the budget is used for RL and the remaining 40% is used for mid-training. Table 4 shows that hybrid mid-training+RL recipes substantially outperform pure RL. Compared with Full RL, the 60% RL mixture improves `op=15-20 pass@128` from 53.2 to 66.6, while the 20% RL mixture gives the strongest `op=11-14` scores and the best `op=15-20 pass@1`. The best allocation therefore varies across metrics at this scale, but the overall conclusion is unchanged: mid-training installs useful priors, while RL further improves exploration and high-$k$ performance. This is the scaled-up counterpart of Section 5,

*Table 3.* Contextual transfer at 500M scale. We report *context B* performance after RL under different amounts of *context B* exposure during pre-training. Contextual transfer requires minimal pre-training exposure: with 0% exposure to *context B*, RL fails completely, while sparse 1–10% exposure already enables strong transfer. Parentheses report absolute change over the 0.1% exposure setting.

| Context-B Exposure | op=2-10 | | op=11-14 | | op=15-20 | |
|---|---|---|---|---|---|---|
| | pass@1 | pass@128 | pass@1 | pass@128 | pass@1 | pass@128 |
| 10% | **82.7** (+7.6) | 95.8 (+10.0) | **41.0** (+6.5) | 71.7 (+20.4) | **39.5** (+20.2) | **54.0** (+2.5) |
| 1% | 82.1 (+7.0) | **98.0** (+12.2) | 39.4 (+4.9) | **76.0** (+24.7) | 31.8 (+12.5) | 51.6 (+0.1) |
| 0.1% | 75.1 | 85.8 | 34.5 | 51.3 | 19.3 | 51.5 |
| 0% | 0.0 (-75.1) | 0.0 (-85.8) | 0.0 (-34.5) | 0.0 (-51.3) | 0.0 (-19.3) | 0.0 (-51.5) |

*Table 4.* Fixed-budget mid-training and RL allocation at 500M scale. Hybrid mid-training+RL recipes outperform Full RL, confirming that the two stages play complementary roles. Parentheses report absolute change over Base for each difficulty range.

| Setting | op=2-10 | | op=11-14 | | op=15-20 | |
|---|---|---|---|---|---|---|
| | pass@1 | pass@128 | pass@1 | pass@128 | pass@1 | pass@128 |
| Base | 96.6 | 99.8 | 47.4 | 77.9 | 14.6 | 45.3 |
| Full RL | 90.3 (-6.3) | 99.5 (-0.3) | 53.4 (+6.0) | 88.6 (+10.7) | 26.2 (+11.6) | 53.2 (+7.9) |
| 80% RL | 90.8 (-5.8) | 99.5 (-0.3) | 83.2 (+35.8) | 98.1 (+20.2) | 31.8 (+17.2) | 66.3 (+21.0) |
| 60% RL | 93.4 (-3.2) | 99.8 (0.0) | 86.5 (+39.1) | 99.1 (+21.2) | 34.9 (+20.3) | **66.6** (+21.3) |
| 40% RL | 95.0 (-1.6) | 99.9 (+0.1) | 88.4 (+41.0) | 98.7 (+20.8) | 34.3 (+19.7) | 62.6 (+17.3) |
| 20% RL | 96.2 (-0.4) | 99.8 (0.0) | **89.9** (+42.5) | **99.6** (+21.7) | **36.4** (+21.8) | 63.5 (+18.2) |

where the main text argues that mid-training and RL should be balanced around these complementary roles.

**Disjoint mid-training and RL distributions.** The previous compute-allocation experiment aligns the target difficulty of mid-training and RL in order to isolate the effect of stage allocation. We additionally test a harder setting where the two stages use different distributions: mid-training uses op=9-12, while RL uses op=11-14. Table 5 shows that the qualitative trend still holds. Even when mid-training and RL are distributionally disjoint, hybrid training improves substantially over the Base model and remains competitive with or stronger than pure RL on the target op=11-14 range. This stress test again supports the conclusion of Section 5: mid-training need not replace RL; it can provide useful priors that RL then exploits.

**Process-aware rewards at 500M scale.** Finally, we evaluate process-aware rewards at 500M scale. Table 6 reports the absolute change over outcome-only rewards under the same RL data distribution. The gains are modest but generally positive, especially on the edge-of-competence regime. For RL on op=11-14, process-aware rewards improve op=11-14 pass@128 by 1.05 and produce positive average gains on OOD metrics. This links back to Section 6: aligning rewards with valid intermediate reasoning can improve reasoning fidelity and reduce reward hacking.

### F.2. Real World Experiments on Qwen2.5-7B

**Setup.** To test whether the same qualitative trends extend beyond synthetic controlled tasks, we conduct additional experiments with Qwen2.5-7B Base. We construct RL training data from a NuminaMath-style math corpus by grouping examples according to the empirical success rate of the base model. Specifically, easy examples are solved greedily by the base model (pass@1 = 1); medium examples are not solved greedily but have non-zero success under a moderate sampling budget (pass@32 $\neq$ 0); and hard examples are not solved within the medium budget but have non-zero success under a larger budget (pass@128 $\neq$ 0). We sample 20K examples for each bucket and run GRPO for 80 steps. For evaluation, we split benchmarks by base-model performance: MATH-500 is the in-domain task, where the base model is already strong (pass@32 $\approx$ 85%), while AIME 2024/2025 are harder out-of-domain tasks with much lower base pass@32.

**RL on edge-of-competence data.** Table 7 shows that the edge-of-competence trend also appears in the 7B setting. On MATH-500, all RL buckets substantially improve pass@1 but do not improve pass@32 over the base model, suggesting that RL mainly sharpens already accessible capabilities on in-domain tasks. On harder AIME benchmarks, especially AIME 2025, edge and hard buckets yield stronger gains. RL on the hard bucket improves AIME 2025 pass@32 from 13.3 to 60.0, indicating that RL can expand capability when the evaluation task leaves sufficient headroom. This real-world result echoes Section 3: RL is most useful when the benchmark is beyond the base model's saturated regime and the training data remain

*Table 5.* Disjoint mid-training and RL distributions at 500M scale. Mid-training uses `op=9-12`, while RL uses `op=11-14`. Hybrid training remains effective even when the two stages are not trained on identical distributions. Parentheses report absolute change over Base for each difficulty range.

| Setting | op=2-10 | | op=11-14 | | op=15-20 | |
|---|---|---|---|---|---|---|
| | pass@1 | pass@128 | pass@1 | pass@128 | pass@1 | pass@128 |
| Base | 96.6 | 99.8 | 47.4 | 77.9 | 14.6 | 45.3 |
| 80% RL | 91.5 (-5.1) | 99.0 (-0.8) | 71.5 (+24.1) | 92.1 (+14.2) | 17.6 (+3.0) | 47.2 (+1.9) |
| 60% RL | 93.6 (-3.0) | 99.5 (-0.3) | 76.0 (+28.6) | 92.8 (+14.9) | 18.7 (+4.1) | **49.3 (+4.0)** |
| 40% RL | 94.4 (-2.2) | 99.5 (-0.3) | **77.2 (+29.8)** | 93.7 (+15.8) | **18.8 (+4.2)** | 49.2 (+3.9) |
| 20% RL | 95.7 (-0.9) | 99.7 (-0.1) | 76.7 (+29.3) | **94.3 (+16.4)** | 16.3 (+1.7) | 46.5 (+1.2) |

*Table 6.* Effect of process-aware rewards at 500M scale. Values denote absolute changes over outcome-only rewards under the same RL data distribution. Positive changes are shown in blue, and negative changes are shown in gray.

| Setting | op=2-10 | | op=11-14 | | op=15-20 | | $\Delta$ID | $\Delta$OOD | $\Delta$Total |
|---|---|---|---|---|---|---|---|---|---|
| | $\Delta$pass@1 | $\Delta$pass@128 | $\Delta$pass@1 | $\Delta$pass@128 | $\Delta$pass@1 | $\Delta$pass@128 | | | |
| RL op=7-10 | -0.00 | +0.10 | +0.06 | -0.44 | +0.01 | +0.00 | +0.02 | +0.04 | +0.03 |
| RL op=9-12 | +0.04 | -0.10 | +0.19 | -0.30 | +0.02 | +0.30 | +0.05 | +0.03 | +0.04 |
| RL op=11-14 | +0.06 | +0.24 | +0.52 | **+1.05** | -0.06 | +0.03 | +0.02 | **+0.21** | **+0.12** |

learnable.

**Fixed-budget allocation between mid-training and RL.** We also study how to allocate a fixed 100-step training budget between mid-training and RL. We compare Base, Full Mid, Full RL, and three hybrid settings with progressively larger RL fractions: Light RL, Medium RL, and Heavy RL. Table 8 shows that mid-training and RL play complementary roles. Full Mid is strong for pass@1, while settings with more RL compute tend to improve high-$k$ performance on harder benchmarks. For example, on AIME 2025, Heavy RL improves pass@32 to 40.0, compared with 33.3 for Full Mid and 26.7 for Full RL. This links back to Section 5: the same fixed-budget trade-off between prior installation and RL exploration also appears in a 7B math setting.

**Cross-task transfer to code and science.** Finally, we test whether lightweight target-domain exposure before RL enables transfer beyond math. Since full pre-training from scratch is infeasible at this scale, we inject a small amount of target-domain data before RL. Specifically, we add either 1% or 5% code/science data during the target-exposure stage, with the remaining data drawn from the math edge-of-competence set. During RL, we use a balanced mixture of math and the target domain. We evaluate code performance on LiveCodeBench v4/v6 and science performance on GPQA Diamond.

Table 9 shows clear transfer to code. With 5% code injection, RL improves LiveCodeBench v4 pass@1 from 0.6 to 6.6 and LiveCodeBench v6 pass@1 from 2.1 to 10.4. It also shows smaller science gains concentrated at low $k$: 5% science exposure followed by RL improves GPQA pass@1 by 4.6 points, while high-$k$ performance is already near saturation. These results support the conclusion that a small amount of target-domain exposure before RL can enable cross-domain transfer, although the strength of the effect depends on the target domain. This is the naturalistic analogue of Section 4: RL can amplify target-domain ability only after the base model has received some exposure to that domain.

**Summary.** Overall, these additional experiments strengthen the external validity of our main claims. The 500M controlled experiments show that the core trends are not restricted to the 100M setting: RL is most effective near the edge of competence, contextual transfer requires non-zero pre-training exposure, mid-training and RL are complementary, and process-aware rewards improve reasoning fidelity. The Qwen2.5-7B experiments further suggest that these trends extend to more realistic math, code, and science settings. At the same time, these results should be interpreted as empirical evidence rather than universal laws: the optimal exposure threshold, difficulty bucket, and mid-training/RL allocation can depend on model scale, data distribution, and target domain.

*Table 7.* **RL data difficulty on Qwen2.5-7B Base.** We group RL data by empirical difficulty under the base model. On in-domain MATH-500, RL mainly improves `pass@1`; on harder OOD tasks, especially AIME 2025, edge-of-competence and hard data produce stronger high-$k$ gains. Parentheses report absolute change over Qwen2.5-7B Base for each benchmark.

| Benchmark | Setting | Performance | | | |
|---|---|---|---|---|---|
| | | pass@1 | pass@4 | pass@8 | pass@32 |
| MATH-500 | Qwen2.5-7B Base | 29.7 | 61.8 | 73.3 | 84.8 |
| | + RL (easy data) | 65.2 (+35.5) | 74.2 (+12.4) | 77.0 (+3.7) | 80.6 (-4.2) |
| | + RL (edge data) | 65.7 (+36.0) | 74.5 (+12.7) | 77.7 (+4.4) | 81.6 (-3.2) |
| | + RL (hard data) | 65.9 (+36.2) | 75.2 (+13.4) | 78.1 (+4.8) | 81.4 (-3.4) |
| AIME 2024 | Qwen2.5-7B Base | 3.1 | 9.3 | 14.4 | 30.0 |
| | + RL (easy data) | 10.5 (+7.4) | 16.5 (+7.2) | 20.9 (+6.5) | 30.0 (0.0) |
| | + RL (edge data) | 10.9 (+7.8) | 17.8 (+8.5) | 21.7 (+7.3) | 30.0 (0.0) |
| | + RL (hard data) | 12.9 (+9.8) | 18.8 (+9.5) | 21.3 (+6.9) | 30.0 (0.0) |
| AIME 2025 | Qwen2.5-7B Base | 0.6 | 2.4 | 4.6 | 13.3 |
| | + RL (easy data) | 4.8 (+4.2) | 13.7 (+11.3) | 19.8 (+15.2) | 33.3 (+20.0) |
| | + RL (edge data) | 11.0 (+10.4) | 21.3 (+18.9) | 25.7 (+21.1) | 33.3 (+20.0) |
| | + RL (hard data) | 11.3 (+10.7) | 27.8 (+25.4) | 38.2 (+33.6) | **60.0** (+46.7) |

*Table 8.* **Fixed-budget trade-off between mid-training and RL on Qwen2.5-7B Base.** We allocate a fixed 100-step budget between mid-training and RL, reported with named allocation settings. Pure mid-training yields the strongest low-$k$ gains, whereas RL-heavy allocation improves higher-$k$ performance, especially on OOD AIME benchmarks. Parentheses report absolute change over Qwen2.5-7B Base for each benchmark.

| Setting | MATH-500 (ID) | | | AIME 2024 (OOD) | | | AIME 2025 (OOD) | | |
|---|---|---|---|---|---|---|---|---|---|
| | pass@1 | pass@4 | pass@32 | pass@1 | pass@4 | pass@32 | pass@1 | pass@4 | pass@32 |
| Qwen2.5-7B Base | 29.7 | 61.8 | 84.8 | 3.1 | 9.3 | 30.0 | 0.6 | 2.4 | 13.3 |
| Full Mid-training | **65.4** (+35.7) | 73.9 (+12.1) | 80.2 (-4.6) | **14.2** (+11.1) | 20.0 (+10.7) | 30.0 (0.0) | **9.8** (+9.2) | 17.5 (+15.1) | 33.3 (+20.0) |
| Light RL | 58.7 (+29.0) | 73.1 (+11.3) | 82.6 (-2.2) | 7.9 (+4.8) | 15.8 (+6.5) | 33.3 (+3.3) | 7.1 (+6.5) | 16.2 (+13.8) | 33.3 (+20.0) |
| Medium RL | 58.7 (+29.0) | 73.4 (+11.6) | **83.6** (-1.2) | 8.0 (+4.9) | 13.1 (+3.8) | 20.0 (-10.0) | 8.3 (+7.7) | 19.9 (+17.5) | 33.3 (+20.0) |
| Heavy RL | 65.2 (+35.5) | **75.9** (+14.1) | 82.2 (-2.6) | 12.7 (+9.6) | **20.7** (+11.4) | **40.0** (+10.0) | 9.0 (+8.4) | **21.1** (+18.7) | **40.0** (+26.7) |
| Full RL | 64.9 (+35.2) | 73.9 (+12.1) | 82.0 (-2.8) | 10.7 (+7.6) | 20.6 (+11.3) | 36.7 (+6.7) | 6.0 (+5.4) | 15.4 (+13.0) | 26.7 (+13.4) |

# G. Training Dynamics for § 3

In this section, we provide a detailed analysis on the training dynamics for different post-training recipes in extrapolative generalization. **NLL Reduction Across Evaluation Ranges.** We analyze the post-training across different post-training data recipes used in § 3 and their impact on NLL reduction across various evaluation operation ranges.

We can observe from Figure 10 that post-training consistently reduces NLL across all evaluation ranges, with the most significant gains occurring in `op=11-14` range. This indicates that the model effectively learns to compose atomic skills to tackle more complex problems. **Post-training Dynamics.** We further investigate the reward dynamics during post-training across different data recipes.

From Figure 11, we observe that post-training on tasks aligned with the model's edge of competence (`op=9-12` and `op=11-14`) leads to significant reward improvements, indicating effective learning. In contrast, when the tasks are too easy (`op=7-10`) or too hard (`op=17-20`), the reward plateaus, suggesting limited learning progress in these regimes.

*Table 9.* **Cross-task transfer from math to code and science on Qwen2.5-7B Base.** Lightweight target-domain exposure alone yields limited gains, but exposure followed by RL substantially improves transfer, especially for code. For science, gains are smaller and concentrated at low $k$. Light blue rows indicate RL applied after the exposure row immediately above. Parentheses report absolute change over the corresponding base model for each benchmark.

| Benchmark | Setting | Performance | | | |
|---|---|---|---|---|---|
| | | pass@1 | pass@4 | pass@8 | pass@16 |
| **Math → Code** | | | | | |
| | Qwen2.5-7B Base | 0.6 | 1.9 | 3.0 | – |
| LiveCodeBench v4 | 1% Code | 1.0 (+0.4) | 3.3 (+1.4) | 5.0 (+2.0) | – |
| | + RL | 4.8 (+4.2) | 10.5 (+8.6) | 13.9 (+10.9) | – |
| | 5% Code | 0.7 (+0.1) | 2.5 (+0.6) | 4.0 (+1.0) | – |
| | + RL | **6.6** (+6.0) | **12.1** (+10.2) | **14.9** (+11.9) | – |
| | Qwen2.5-7B Base | 2.1 | 6.9 | 10.3 | – |
| LiveCodeBench v6 | 1% Code | 2.4 (+0.3) | 7.0 (+0.1) | 9.7 (-0.6) | – |
| | + RL | 9.7 (+7.6) | 15.4 (+8.5) | 17.7 (+7.4) | – |
| | 5% Code | 2.4 (+0.3) | 7.5 (+0.6) | 11.4 (+1.1) | – |
| | + RL | **10.4** (+8.3) | **17.4** (+10.5) | **20.0** (+9.7) | – |
| **Math → Science** | | | | | |
| | Qwen2.5-7B Base | 27.8 | 69.4 | 88.5 | 98.0 |
| GPQA Diamond | 1% Science | 29.7 (+1.9) | 61.7 (-7.7) | 75.7 (-12.8) | 91.9 (-6.1) |
| | + RL | 29.0 (+1.2) | 67.7 (-1.7) | 84.4 (-4.1) | 96.0 (-2.0) |
| | 5% Science | 31.1 (+3.3) | 67.3 (-2.1) | 84.4 (-4.1) | 98.5 (+0.5) |
| | + RL | **32.4** (+4.6) | **70.0** (+0.6) | 85.6 (-2.9) | 98.0 (0.0) |

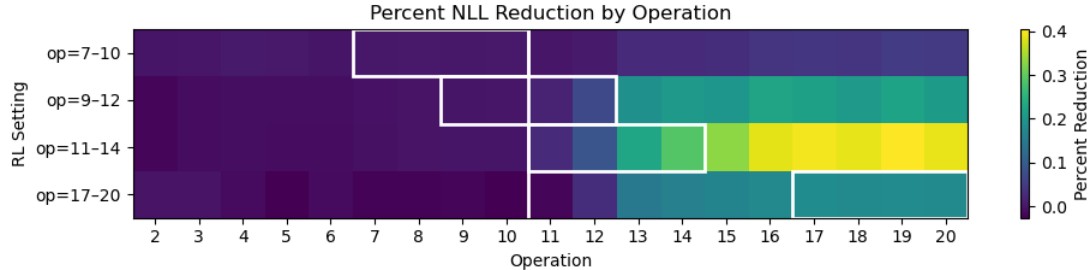

*Figure 10.* **NLL reduction compared with the base model.** White boxes denote RL-trained operation ranges. NLL gains decay smoothly as the evaluation range diverges from the RL-trained operations. Notably, RL on `op=11-14` achieves the largest NLL reduction on `op=15-20`.

## H. Detailed Analysis of Post-Training Effects on Contextual Generalization

In this section, we provide a detailed analysis of how different post-training data recipes affect contextual generalization to long-tailed contexts given atomic reasoning primitives during pre-training.

### H.1. When Reasoning Primitives are Shared During Pre-Training

Beyond mastering fundamental reasoning skills, an essential dimension of model generalization lies in *contextual generalization*—the capacity to transfer learned reasoning behaviors across diverse problem contexts, such as varying surface narratives or domains. In this section, we investigate whether post-training can incentivize models to generalize reasoning competence to *long-tailed* or underrepresented contexts that were scarcely observed during pre-training.

**Task Settting.** We study two distinct problem contexts: a frequent, canonical *context A* and a long-tailed *context B*, both sharing the same underlying reasoning priors (logical-arithmetic reasoning in our case, detailed context settings can be found in Appendix K). The pre-training corpus consists of 99.9% *context A* and only 0.1% *context B*, both spanning `op=2-20`. During post-training, we vary the exposure to *context B* across 200K samples with different ratios: 0%, 2%, 10%, 50%, and

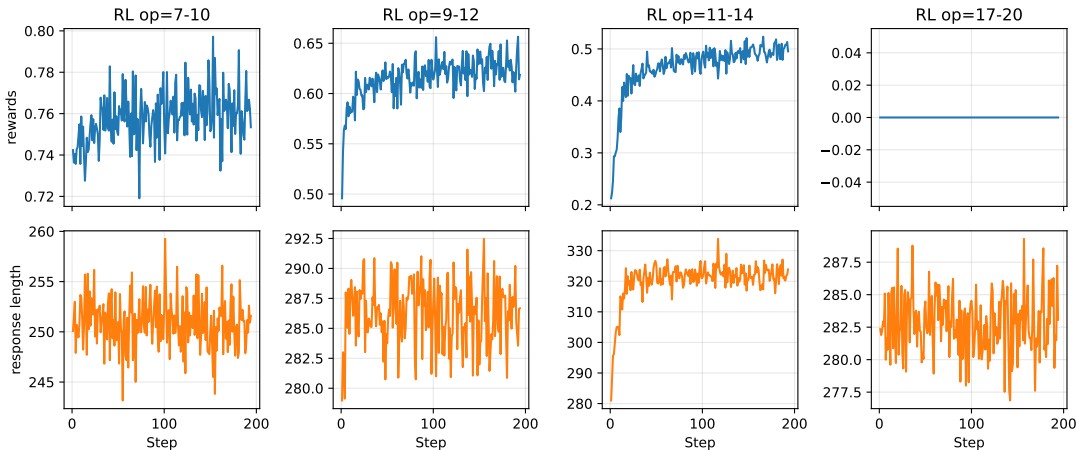

*Figure 11.* **Reward dynamics across different post-training data recipes.** RL on `op=9-12` and `op=11-14` tasks, which are calibrated to the model's *edge of competence*, leads to genuine improvements in reasoning. However, when the task difficulty is either too easy or too hard, the reward stagnates, indicating limited learning progress.

100%.

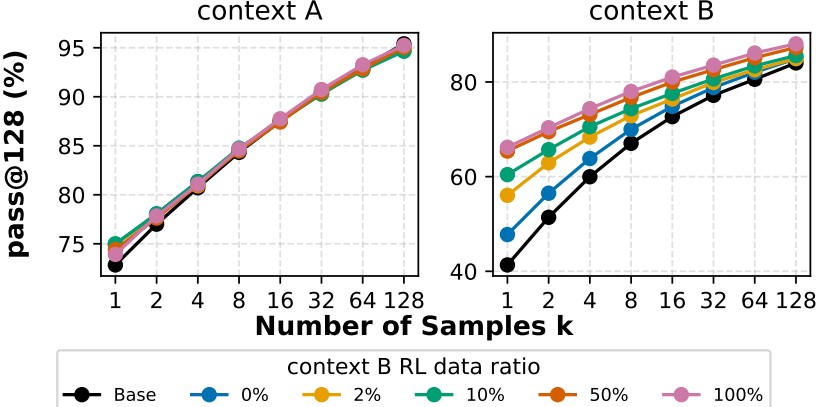

*Figure 12.* `pass@k` performance on contextual generalization tasks after post-training with varying exposure to *context B*. With shared reasoning primitives during pre-training, models exhibit strong transfer to *context B* even with limited or no exposure during post-training.

---

**Observation 5**

With shared reasoning primitives during pre-training, there is a positive relation between exposure to *context B* during post-training and performance on *context B*. Notably, even without any *context B* exposure during post-training, the model still achieves significant transfer, underscoring the role of shared primitives in enabling contextual generalization.

---

**Takeaway 5**

**When atomic primitives are shared, post-training can incentivize generalization to long-tailed contexts.** Remarkably, even with a 0% exposure to *context B* during post-training, the model achieves substantial transfer, highlighting the critical role of shared reasoning structures during pre-training.

---

## H.2. When Only Atomic Primitives are Exposed During Pre-Training

We next examine contextual generalization when the base model has only been exposed to *basic atomic primitives* in the long-tailed context during pre-training.

**Task Setting.** With the same contextual data distribution as above, we restrict *context B* data during pre-training to only atomic operations, while *context A* spans the full range. The pre-training corpus consists of 99% *context A* (`op=2-20`) and only 1% *context B*, with *context B* restricted to atomic operations (`op=2`). Thus, **the model learns reasoning structures primarily through *context A*, while having minimal exposure to the surface forms of *context B*.** During post-training, we perform RL fine-tuning with 200K samples where the ratio of *context B* data varies across five regimes: 0%, 1%, 10%, 50%, and 100%. Detailed data recipes can be found in Appendix K.

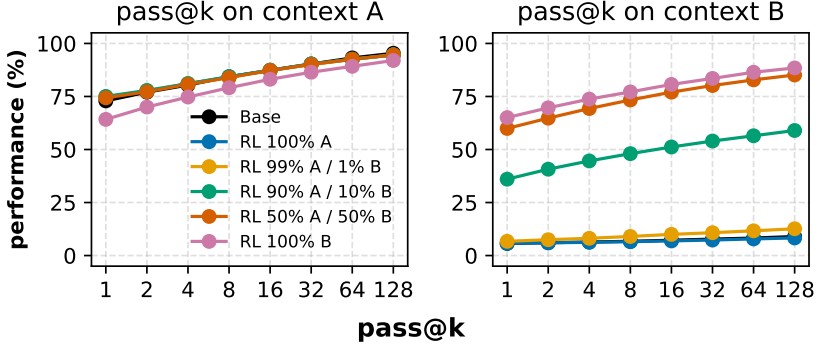

*Figure 13.* `pass@k` performance for different contexts with base model limited to basic atoms for *context B*. Post-training on *context A* maintains stable performance, while exposure of 10% *context B* during RL enables contextual transfer.

---

**Observation 6**

As shown in Figure 13, post-training exclusively on *context A* or with only *extremely sparse exposure* to *context B* (0–1%) maintains strong performance within *context A* but yields minimal transfer to the long-tailed *context B*. However, once a small amount of *context B* data is introduced—around 10% of total samples—*context B* performance improves dramatically, with `pass@128` accuracy increasing by over +76 points. Further increasing the proportion of *context B* data (50%, 100%) brings diminishing gains, indicating that RL rapidly establishes robust cross-context reasoning once minimal supervision is available. Notably, even when post-training uses *100% context B* data—entirely distinct from the dominant pre-training context—*context A* performance remains stable. This shows that RL enables model to learn transferable reasoning policies that extend across surface forms while preserving competence in previously mastered contexts.

---

**Takeaway 6**

**RL enables stable cross-context generalization under extreme imbalance.** Even when the base model has only minimal exposure to long-tailed contexts during pre-training, RL fine-tuning can transfer reasoning competence across domains by leveraging shared reasoning structures.

---

## H.3. Training Dynamics for § H.2

We plot the post-training reward dynamics across different data recipes used in § H.2 to further understand how varying exposure to long-tailed contexts during RL affects learning progress.

From Figure 14, we can observe that when the exposure to *context B* during post-training is extremely limited (0-1%), the reward plateaus, indicating minimal learning progress. However, with moderate exposure (10-100%), the reward improves significantly, reflecting effective learning and transfer to the long-tailed context.

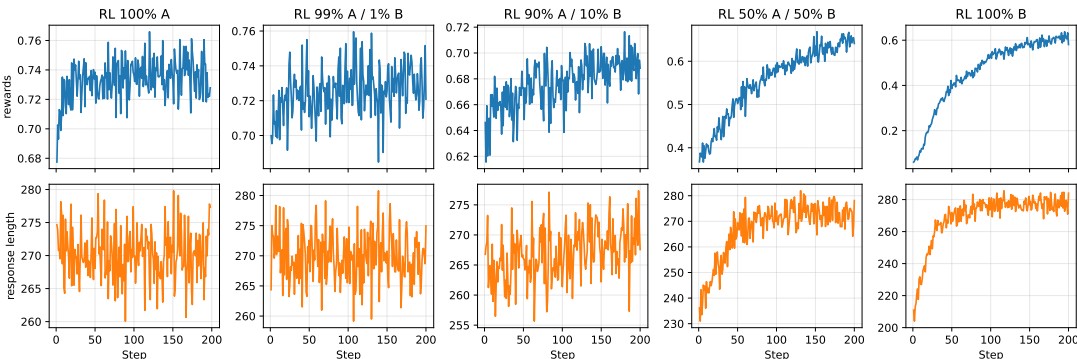

*Figure 14.* Reward dynamics across different post-training data recipes. When RL exposure to *context B* is extremely limited (0-1%), the reward stagnates. However, with moderate exposure (10-100%), the reward improves significantly, reflecting effective learning and transfer.

# I. Detailed Analysis of Pre-Training Effects on Extrapolative Generalization

Pre-training defines the atomic reasoning primitives that post-training can later compose and extend. If the base model already encounters moderately complex problems during pre-training, post-training may push those primitives toward deeper, compositional reasoning. Otherwise, post-training may lack the scaffolding to explore beyond its inherited competence. We thus study how varying pre-training difficulty influences subsequent extrapolative generalization.

**Task Setting.** We fix the post-training recipe to 200K samples from the `op=11-14` range, previously identified as a *edge of competence* (see Figure 3). We then vary the proportion of "hard" data (`op=7-10`) included during pre-training to assess how exposure to complex primitives affects the base model's ability to generalize after RL. (See Appendix K for detailed data recipes.)

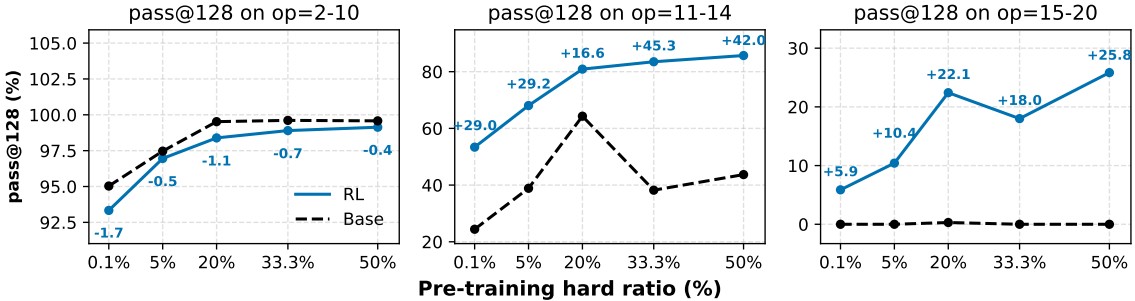

*Figure 15.* `pass@128` performance on extrapolative tasks after post-training on `op=11-14`, under varying levels of hard-data exposure during pre-training.

---

**Observation 7**

As shown in Figure 15, greater exposure to hard problems during pre-training consistently improves both base and post-trained performance. However, the *marginal gain from RL* diminishes as pre-training becomes more comprehensive. When pre-training already covers a substantial fraction of mid-depth tasks, RL adds only modest improvement. By contrast, when pre-training includes limited but nontrivial exposure to difficult primitives (e.g., 20% of `op=7-10`), RL produces the largest relative boost—enhancing `pass@128` accuracy on `op=15-20` by more than +22 points. This suggests that RL is most effective when the model's prior competence is partial—strong enough to support exploration, but incomplete enough to leave room for discovery.

---

> **Takeaway 7**
>
> **Pre-training establishes the foundation, RL extends it.** Rich exposure to compositional primitives during pre-training enables RL to push reasoning depth beyond the pre-training range. Yet the benefits of RL taper off once those primitives are fully mastered, highlighting the complementary roles of the two stages.

### I.1. Training Dynamics for § I

We analyze the training dynamics during post-training across different pre-training data recipes.

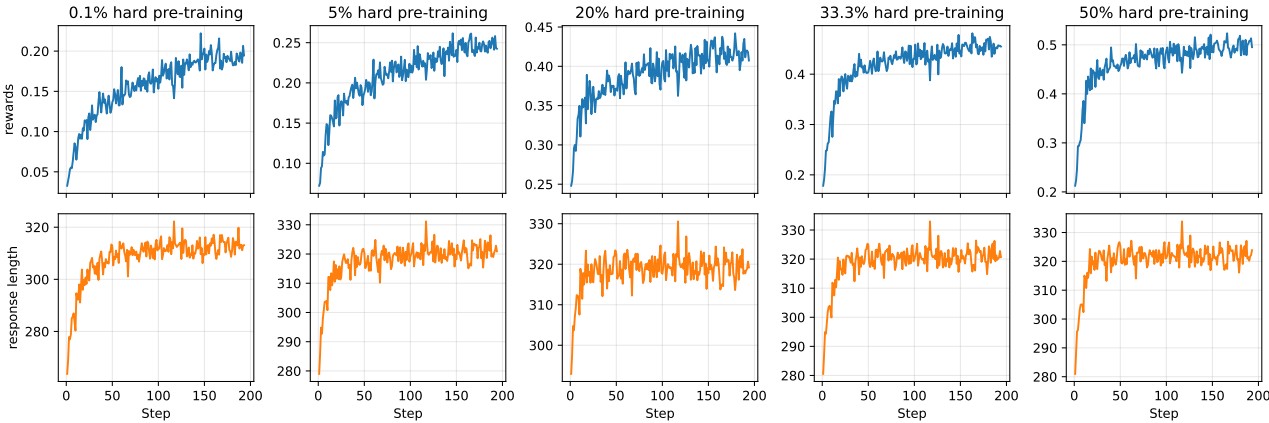

*Figure 16.* Reward dynamics across different pre-training data recipes. Models with moderate hard-data exposure (20-50%) during pre-training exhibit significant reward improvements during post-training, indicating effective learning and extrapolation. In contrast, models with either too little (0%) or too much (100%) hard-data exposure show limited reward gains, suggesting constrained learning progress.

## J. Training Dynamics for § 4

In this section, we provide an analysis of the training dynamics for different pre-training data recipes in contextual generalization in § 3. From Figure 17, we observe that moderate exposure ratio to long-tailed contexts, even with basic

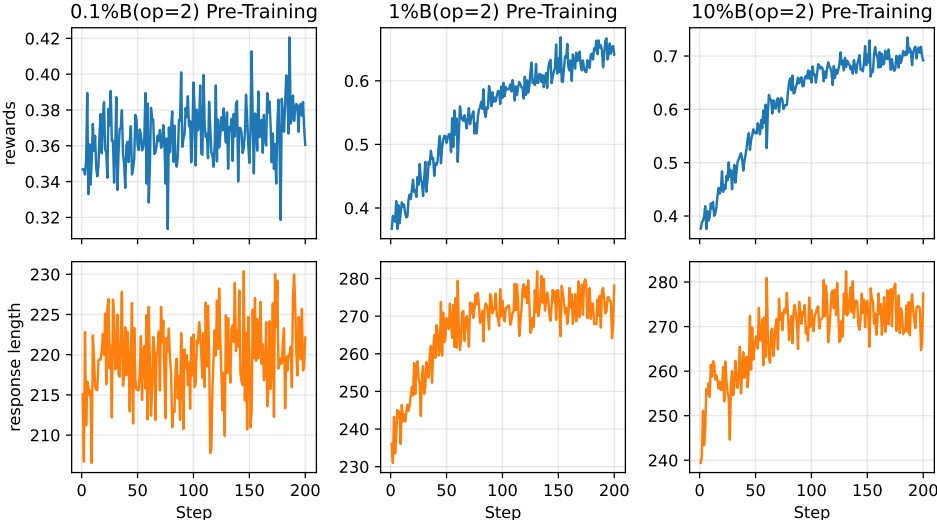

*Figure 17.* Reward dynamics across different pre-training data recipes. Models with minimal exposure to long-tailed contexts exhibit no reward improvement during post-training. While models with moderate to full exposure show significant reward improvements, indicating effective learning and contextual generalization.

primitives during pre-training, is necessary for the model to make significant reward improvements during post-training.

# K. Post-Training and Pre-Training Data Recipe

In this section, we detail the data recipes employed in § 3 § 4, § H.1, § H.2, and § I. Table 10 summarizes the specific operation count ranges, contextual templates, and training budgets utilized across different experimental sections.

| Section | Pre-training | | | Post-training (RL) | | |
| | op($\mathcal{G}$) | Contexts | Training Budget | op($\mathcal{G}$) | Contexts | Training Budget |
|---|---|---|---|---|---|---|
| § 3 | $20\%$op=2−4 + $30\%$op=5−7 + $50\%$op=8−10 | 33%A+33%B+33%C | 10B tokens | op=8−10
op=9−12
op=11−14
op=17−20 | 33%A+33%B+33%C | 204.8k samples |
| § 4 | $100\%$op=2−20 A + $0\%$op=2 B
$99.9\%$op=2−20 A + $0.1\%$op=2 B
$99\%$op=2−20 A + $1\%$op=2 B
$90\%$op=2−20 A + $10\%$op=2 B | | 10B tokens | op=2−20 | 50% A + 50% B | 204.8k samples |
| § H.1 | op=2−20 | 99.9%A+0.1%B | 10B tokens | op=2−20 | 100% A
98%A + 2%B
90%A + 10%B
50%A + 50%B
100%B | 204.8k samples |
| § H.2 | $99\%$op=2−20 A + $1\%$op=2 B | | 10B tokens | op=2−20 | 100% A
99%A + 1%B
90%A + 10%B
50%A + 50%B
100%B | 204.8k samples |
| § I | $99.9\%$ op=2−6 + $0.1\%$ op=8−20
$49.95\%$ op=2−4 + $49.95\%$ op=5−7 + $0.1\%$ op=8−10
$47.5\%$ op=2−4 + $47.5\%$ op=5−7 + $5\%$ op=8−10
$50\%$ op=2−4 + $30\%$ op=5−7 + $20\%$ op=8−10
$20\%$ op=2−4 + $30\%$ op=5−7 + $50\%$ op=8−10 | 33%A+33%B+33%C | 10B tokens | op=11−14 | 33%A+33%B+33%C | 204.8k samples |

*Table 10.* Data recipes for pre-/post-training experiments in § 3, § 4, § H.1, § H.2, and § I. op($\mathcal{G}$) ranges indicate the operation counts during each training phase. Contexts A, B, C correspond to distinct templates: A = *animals–zoo*, B = *teachers–school*, C = *movie-festival*. The data recipes for different operation ranges and contexts are uniformly sampled within the specified proportions. Shaded cells indicate the ablated settings.

# L. Mid-/Post-Training Mixing with Different Computation Budget

In this section, we first detail the compute budget formulation for mid-training and RL equivalence, then provide the exact data recipes for combining mid-training and post-training under different total compute budgets.

## L.1. Compute Budget of Mid-Training and RL Equivalence

**Training Computation.** Following the Chinchilla scaling law (Hoffmann et al., 2022), a decoder-only Transformer with $P$ non-embedding parameters trained on $T$ tokens consumes approximately

$$C_{\text{train}} \approx 6PT \quad flops. \tag{4}$$

Thus, a mid-training phase with budget $T_{\text{mid}}$ incurs $C_{\text{mid}} = 6PT_{\text{mid}} \quad flops$.

**Fine-Grained RL Computation.** For on-policy GRPO, computation can be decomposed as:

- **Rollout:** actor model forward ($2P$),

- **Reference (optional):** reference model forward ($2P$),

- **Policy Update:** forward ($2P$) and backward ($4P$) passes.

Summing these terms yields:
$$C_{\text{RL}} = (8 + 2\gamma)P\,N\,r\,L_{\text{total}}, \tag{5}$$

where $\gamma \in \{0, 1\}$ toggles the reference-model pass, $N$ is the number of RL samples, $r$ is the rollout size, and $L_{\text{total}}$ is the total sequence length (including both prompt and completion).

**Mid-training Token Equivalence.** Normalizing by Equation 4 gives the equivalent mid-training token cost:

$$T_{\text{RL}} = \frac{C_{\text{RL}}}{6P} = \left(\tfrac{4}{3} + \tfrac{\gamma}{3}\right)NrL_{\text{total}}. \tag{6}$$

When $\gamma = 1$, we obtain the equivalence used in the main text:

$$\boxed{T_{\text{RL}} = \tfrac{5}{3}NrL_{\text{total}}.}$$

**Budget Allocation and Step Calculation.** Given total budget $T$ and RL ratio $\beta$,

$$T_{\text{mid}} = (1 - \beta) \cdot T, \qquad\qquad T_{\text{RL,eq}} = \beta \cdot T. \tag{7}$$

The corresponding number of RL samples $N(p)$ and update steps are:

$$N(\beta) = \frac{3}{5} \cdot \frac{\beta T}{rL_{\text{total}}}, \qquad\qquad \text{steps}_{\text{RL}}(p) = \frac{N(\beta)}{B}, \tag{8}$$

where $r = 6$ is the rollout size, $L_{\text{total}} = 2048$ is the total sequence length, $B = 1024$ is the RL batch size, and $T$ is the total token budget. The mid-training steps are:

$$\text{steps}_{\text{mid}}(\beta) = \frac{T_{\text{mid}}}{B_{\text{mid}} \cdot L_{\text{mid}}}, \tag{9}$$

where $B_{\text{mid}} = 512 \times 1024$ is the mid-training batch size and $L_{\text{mid}} = 2048$ is the mid-training sequence length.

**Task Setting.** We use 10B tokens with 20% `op=2-4`, 30% `op=5-7`, and 50% `op=8-10` for pre-training. To avoid catastrophic forgetting during mid-training, we use 20% budget for `op=2-10` and 80% for `op=11-14` during mid-training. For fair comparison, RL is performed with the same data distribution as mid-training. Table 11 details the exact step counts for mid-training and RL across varying total token budgets $T$ and mid-training ratios $p$. We perform mid-/post-training with **Full mid-training**, **Full RL**, **Light-RL** ($\beta = 0.2$), **Medium-RL** ($\beta = 0.5$), and **Heavy-RL** ($\beta = 0.8$) under different total compute budgets.

*Figure 18.* `pass@k` performance for different mid-training and RL mixing ratios under varying total compute budgets.

| Total (B) | Mid-Only steps$_{mid}$ | RL-Only steps$_{RL}$ | Samp.(k) | 80% Mid / 20% RL steps$_{mid}$ | steps$_{RL}$ | 50% Mid / 50% RL steps$_{mid}$ | steps$_{RL}$ | 20% Mid / 80% RL steps$_{mid}$ | steps$_{RL}$ |
|---|---|---|---|---|---|---|---|---|---|
| 1.05 | 2,000 | 50 | 51.2 | 1,600 | 10 | 1,000 | 25 | 400 | 40 |
| 2.10 | 4,000 | 100 | 102.4 | 3,200 | 20 | 2,000 | 50 | 800 | 80 |
| 4.20 | 8,000 | 200 | 204.8 | 6,400 | 40 | 4,000 | 100 | 1,600 | 160 |
| 8.40 | 16,000 | 400 | 409.6 | 12,800 | 80 | 8,000 | 200 | 3,200 | 320 |
| 12.58 | 24,000 | 600 | 614.4 | 19,200 | 120 | 12,000 | 300 | 4,800 | 480 |
| 16.78 | 32,000 | 800 | 819.2 | 25,600 | 160 | 16,000 | 400 | 6,400 | 640 |
| 20.00 | 38,147 | 954 | 976.6 | 30,517 | 191 | 19,073 | 477 | 7,629 | 763 |

*Table 11.* Experimental configurations across varying compute budget scales. We fix the mid-training batch size at 512K tokens. The table maps the total token budget $T$ to the specific step counts required for pure mid-training ($p = 1.0$), pure RL ($p = 0.0$), and hybrid splits.

**Observation 8**

As shown in Figure 18, across all compute budgets *Light-RL* achieves the best OOD-edge `pass@1`. While *Heavy-RL* consistently attains the highest OOD-hard `pass@1` performance. For `pass@128`, when the compute budget is limited (4.2B tokens), *Heavy-RL* achieves the best performance in the OOD-hard setting. When the budget increases (8.4B tokens and above), *Full RL* attains the highest OOD-hard `pass@128` performance.

**Takeaway 8**

**Mid-training and post-training complement each other across varying compute budgets.** A combination of mid-training and RL post-training consistently outperforms either approach individually for `pass@1` tasks. For `pass@128`, the optimal post-training allocation depends on the available compute budget: with limited resources, allocating around 80% to RL strikes a balance between stability and exploration, while with more compute, full RL maximizes extrapolative gains.

