# OpenReview forum: "On the Interplay of Pre-Training, Mid-Training, and RL on Reasoning Language Models"
_ICML.cc/2026/Conference — ICML 2026 spotlight_

### Official Review · Reviewer_qnHT · 2026-02-25

**Soundness:** 4
**Presentation:** 4
**Significance:** 3
**Originality:** 4
**Overall Recommendation:** 5
**Confidence:** 4

**Summary:**

This work conducts well-controlled experiments to investigate the interplay between pre-training, mid-training, and reinforcement learning (RL). Their training data is based on GSM-Infinity, a framework that generates reasoning chains of varying depths by extending dependency graphs.

The experiments show that RL leads to genuine improvements in model capability—measured by potential performance metrics such as pass@128—only when two conditions are met: (1) the pre-training phase leaves sufficient headroom for further improvement, and (2) the RL training data is positioned at the model’s “edge of competence.”

**Compliance With Llm Reviewing Policy:**

Affirmed.

**Key Questions For Authors:**

1. The paper uses 100M-parameter decoder-only Qwen2.5 as the base model for experiments, including pretraining, mid-training and RL. Does it mean that this base model is randomly initialized? (Only the structure is the same as Qwen2.5).

**Limitations:**

yes

**Strengths And Weaknesses:**

Strengths:
1. The presentation is great and easy to follow.
2. The authors wisely choose GSM-infinity to generate data for training and evaluation, whose data complexity is well defined by the number of operations on the computational graph. This makes it possible to study the reasoning capability in a controllable setting. Making the findings more reliable.

Weaknesses:
1. Some of the findings are not suprising and have been well verified in other studies, like: 1) RL data should be around the model’s edge of competence; 2) mid-training (for injecting priors) and RL-training (scaling exploration) should be balanced. 3) We should combine sparse outcome signals with dense process-level feedback.
2. While experimental setting is in a well controlled setting, the training and evaluation data only covers high school level math reasoning data, which raises a question that are the findings applicable to other real-world agentic tasks?
3. In the experiments, different semantic template are applied to the same computational graph to synthesis data. This could introduce redundancy and these data may not be well proper to study the contextual generalization of the model.

---

> ### Author Rebuttal · Authors · 2026-03-31
>
> Thanks for your time and effort. Here's our response:
>
> > W1. **Some of the findings are not surprising and have been well verified in other studies.**
>
> We appreciate this comment and broadly agree that several high-level takeaways are directionally consistent with prior work. Our goal is therefore not to claim that each individual observation is surprising in isolation, but to provide a **controlled framework** that isolates the interactions among **pre-training, mid-training, and RL** under explicit control of task structure, context, and reward design.
>
> In the rebuttal we also extend the empirical scope with **larger-scale models and preliminary real-world evaluations** (see `responses to W1 of reviewers MQpJ and tWW`), which show trends consistent with the conclusions reported in the paper. In the revision, we will clarify this positioning and present the framework as a **foundation for future work** studying these stage interactions in broader and more naturalistic settings.
>
> > W2. The training and evaluation data only cover high school level math reasoning data; are the findings applicable to other real-world agentic tasks?
>
> Thank you for raising this point. Agentic tasks typically involve **environment interaction and stochastic feedback**, which introduce additional sources of uncertainty (e.g., tool outcomes, environment state changes). Studying such interaction-driven effects is outside the scope of our current controlled framework, whose goal is to isolate the interplay between **pre-training, mid-training, and RL** under process-verifiable reasoning settings.
>
> That said, we agree that testing the conclusions beyond synthetic arithmetic tasks is important. As noted in our response to **W1**, we are currently extending the experiments to more **naturalistic reasoning tasks**, including **coding and science domains** (see `response to MQpJ W1`), to examine whether the same trends hold in broader real-world settings.
>
> We will clarify this scope explicitly in the revision and discuss it in the **limitations** section.
>
> > W3. Different semantic templates are applied to the same computational graph, which may introduce redundancy and may not be proper for studying contextual generalization.
>
> We understand the concern, but this design is intentional. The reason we hold the underlying computation graph fixed while varying the surface template is to isolate **contextual transfer** from **structural generalization**. If both the graph structure and the surface form changed simultaneously, it would be difficult to determine whether failures arise from context shift or reasoning shift.
>
> Importantly, during data generation, we **do not instantiate multiple templates for the same task instance**. Instead, each example is generated by **randomly sampling a template and independently sampling a computational graph**, so different templates do not systematically replicate the same task. In addition, we perform **structural deduplication** over generated graphs to remove identical reasoning structures.
>
> We will clarify this generation procedure and the deduplication process more explicitly in the revision. The resulting setup is therefore designed to study transfer across **contextual realizations of the same reasoning primitives**, rather than broader structural generalization.
>
> > Q1. Does using 100M-parameter decoder-only Qwen2.5 mean the base model is randomly initialized?
>
> Yes. We use the Qwen2.5 decoder-only architecture as the backbone specification, but the 100M model in our experiments is trained from scratch on our synthetic corpus rather than initialized from a pretrained Qwen checkpoint. We will make this explicit earlier in the experimental setup to avoid ambiguity.

---

> > ### Author Rebuttal · Reviewer_qnHT · 2026-04-01
> >
> > The authors' responses make sense to me and this is a great paper with serious experimental design. I've raised the score to 5.

---

> > > ### Author Response · Authors · 2026-04-08
> > >
> > > Thanks again for the positive feedback.
> > >
> > > We have also added additional **scaling experiments** in our response to `Reviewer tWWJ`, extending the controlled setup to a **500M Qwen2.5 model**, where we observe trends consistent with our main conclusions.
> > >
> > > In addition, our response to `Reviewer MQpJ` includes **real-world experiments on Qwen2.5-7B Base**, covering (i) RL on edge-of-competence data, (ii) mid-training vs. RL compute allocation under fixed budget, and (iii) cross-task transfer to code (LiveCodeBench) and science (GPQA).
> > >
> > > If interested, please feel free to take a look there.

---

### Official Review · Reviewer_AVFA · 2026-03-13

**Soundness:** 4
**Presentation:** 4
**Significance:** 4
**Originality:** 4
**Overall Recommendation:** 6
**Confidence:** 4

**Summary:**

This work presents a comprehensive investigation of the interplay between pre-training, mid-training and RL for reasoning models on a suit of synthetic tasks/scenarios suitable for research investigation. Rich experiments are conducted, which produces conclusions closely matching observations in recent works on reasoning models and reinforcement learning. Several main takeaways are concluded, including that RL yields genuine capability gains only at the model's edge of competence, that contextual generalization requires minimal but sufficient pre-training exposure (≥1%), that mid-training provides substantial gains under fixed compute, and that process rewards mitigate reward hacking. The authors also provide practical guidance for larger-scale experiments and research.

**Compliance With Llm Reviewing Policy:**

Affirmed.

**Key Questions For Authors:**

1. Do the conclusions made still hold in large scale models?
2. Would it be important to use a disjoint distribution between pre/mid-training and RL?
3. In real-world tasks, different domains usually have very different reasoning structure. Do authors expect ≥1% exposure threshold to still hold when the target domain requires not just surface adaptation but challenging recombination of primitives in novel ways?

**Limitations:**

See questions and weakness.

**Strengths And Weaknesses:**

Strength:
- This paper provides strong evidences that RL can improve the generalization ability. RL on in-distribution tasks just refines what the model already knows, while RL at the edge of competence produces real new capabilities.
- The controlled experiment design enables careful measurement of the functionality of each components in pre-training, mid-training, and post-training.
- This paper highlights mid-training as a rarely explored area and shows that mid-training with even a small amount of compute can provide meaningful gains.

Weakness:
- The experiments are conducted on small-scale models and synthetic tasks only. It would be better if the authors also conduct experiment on a large-scale model to verify effectiveness of their conclusions, especially the mid-training compute vs. RL compute argument.

---

> ### Author Rebuttal · Authors · 2026-03-31
>
> Thanks for your insightful feedback. Here's our response:
>
> > W1. The experiments are conducted on small-scale models and synthetic tasks only.
>
> Please refer to our response to `reviewer Mqpj W1 & tWWJ W1` for real-world tasks and scalability of the conclusion.
>
> > Q1.  Results on large-scale models?
>
> We conducted experiments on the Qwen 2.5 500M model in `response to reviewer tWWJ W1`. We observed similar trends as in 100M model.
>
> > Q2. The data distribution between pre/mid-training and RL?
>
> This is an important point, and we will clarify the distinction more carefully in the revision.
>
> At the instance level, we already use **disjoint partitions across pre-training, mid-training, and post-training** to avoid contamination. At the distributional level, however, Section 5 intentionally keeps the **target difficulty of mid-training and RL aligned**, because the goal of that experiment is to study how a fixed compute budget should be allocated between **installing priors (mid-training)** and **scaling exploration (RL)** under the same target regime.
>
> This design allows us to **attribute performance differences to stage allocation**, rather than introducing an additional confound from **distribution shift**. We agree that a fully distributionally disjoint mid-training/RL setup is an important extension, and we will state this more explicitly as future work.
>
> > Q3. ≥1% exposure threshold in real-world domains
>
> Our current experiments are conducted in a controlled setting where different contexts share the same underlying reasoning graphs and mainly differ in surface semantics. In this regime, we observe that a small but non-zero exposure (≈1%) is sufficient for contextual transfer to reliably emerge.
>
> For real-world domains, however, reasoning structures can differ substantially and may require more complex recombination of primitives. In such settings, we do **not expect the ≈1% threshold to be a stable or universal constant**. Rather, it should be interpreted as an empirical observation in our controlled setup.
>
> We are currently running additional experiments on more naturalistic reasoning tasks to further investigate this question and will include these results in the rebuttal update. We will also clarify this point in the revision and explicitly discuss it in the **limitations** section.

---

> > ### Author Rebuttal · Reviewer_AVFA · 2026-04-03
> >
> > My questions are resolved by the author response. I will maintain my score.

---

> > > ### Author Response · Authors · 2026-04-08
> > >
> > > Thanks again for the positive feedback.
> > >
> > > We have also added additional **scaling experiments** in our response to `Reviewer tWWJ`, extending the controlled setup to a **500M Qwen2.5 model**, where we observe trends consistent with our main conclusions.
> > >
> > > In addition, our response to `Reviewer MQpJ` includes **real-world experiments on Qwen2.5-7B Base**, covering (i) RL on edge-of-competence data, (ii) mid-training vs. RL compute allocation under fixed budget, and (iii) cross-task transfer to code (LiveCodeBench) and science (GPQA)
> > >
> > > If interested, please feel free to take a look there.

---

### Official Review · Reviewer_tWWJ · 2026-03-13

**Soundness:** 4
**Presentation:** 4
**Significance:** 4
**Originality:** 3
**Overall Recommendation:** 5
**Confidence:** 4

**Summary:**

This paper studies the interaction between pre-training, mid-training, and RL post-training for reasoning language models using a controlled synthetic framework based on GSM-Infinite-style tasks, explicit dependency graphs, and process-verified evaluation. The main empirical claims are that RL improves pass@128 beyond the base model mainly when the RL data sit near the model’s edge of competence, that contextual transfer requires at least sparse pre-training exposure to the target context, that mid-training can improve the compute-performance tradeoff relative to RL alone, and that process-aware rewards reduce reward hacking and improve accuracy on harder extrapolative settings.

**Compliance With Llm Reviewing Policy:**

Affirmed.

**Final Justification:**

Overall, the rebuttal successfully addresses my main concerns, and I therefore maintain my positive score.

**Key Questions For Authors:**

- How sensitive are the main conclusions to model scale and base architecture?

- How robust are the headline results to alternative evaluation criteria, such as final-answer-only pass@k or weaker process checks?

- For the contextual-transfer result, can the authors show whether the 1% threshold is stable across templates, seeds, and different definitions of “atomic” exposure?

**Limitations:**

The paper does not adequately discuss limitations or negative societal impact. The current statement is too generic and does not engage with the paper’s actual scope or risks.
A better version should explicitly note the narrow empirical scope, namely a 100M model, synthetic arithmetic-like tasks, and process-based evaluation, and should discuss possible risks of overgeneralizing these findings to broader reasoning settings.

**Strengths And Weaknesses:**

**Strengths**

* This work provides a refreshingly clean empirical analysis of the interplay between pre-training, mid-training, and RL. By isolating these stages in a controlled environment, the authors offer insights into the learning dynamics that are often obscured in larger-scale, end-to-end training reports.
* The experimental design is logically structured, effectively decoupling the specific contributions of RL beyond pre-training and examining the emergence of contextual transfer under fixed compute budgets.
* The paper offers tangible takeaways for the community, particularly the "edge-of-competence" hypothesis and the practical trade-offs between mid-training and RL.

**Weaknesses**

* A primary concern is the disconnect between the broad framing of "reasoning LLMs" and the narrow scope of the experiments. While the paper makes ambitious claims, the evidence relies almost exclusively on a 100M-parameter model fine-tuned on synthetic, DAG-structured arithmetic tasks. It is unclear whether these findings generalize to broader reasoning capabilities or larger-scale models.

* The core conclusions depend heavily on the pass@128 metric using a strict process parser. However, the manuscript lacks a necessary sensitivity analysis regarding parser errors or formatting variations.

---

> ### Author Rebuttal · Authors · 2026-03-31
>
> > W1.  … It is unclear whether these findings generalize to broader reasoning capabilities or larger-scale models.
>
> Thanks for your insights. We've extended our setting to 500M qwen 2.5 model with the same training setup in our paper.
> Here's our results:
>
> ### RL on edge data:
> | Model | op2–10 p@1 | op2–10 p@128 | op11–14 p@1 | op11–14 p@128 | op15–20 p@1 | op15–20 p@128 |
> |---|---:|---:|---:|---:|---:|---:|
> | PT base | 96.6% | 99.8% | 47.4% | 77.9% | 14.6% | 45.3% |
> | RL op7–10 | 90.6% | 99.5% | 40.3% | 75.0% | 16.0% | 48.9% |
> | RL op9–12 | 90.4% | 99.6% | 53.0% | 88.6% | 23.7% | 49.6% |
> | RL op11–14 | 90.3% | 99.5% | 53.4% | 88.6% | 26.2% | **53.2%** |
> | RL op17–20 | 90.3% | 99.6% | 40.6% | 76.3% | 16.3% | 49.3% |
>
> ### RL on different context B context exposure, performance on context B.
> | Model | op2–10 p@1 | op2–10 p@128 | op11–14 p@1 | op11–14 p@128 | op15–20 p@1 | op15–20 p@128 |
> |---|---:|---:|---:|---:|---:|---:|
> | 10% context B | 82.7% | 95.8% | 41.0% | 71.7% | 54.0% | 39.5% |
> | 1% context B | 82.1% | 98.0% | 39.4% | 76.0% | 51.6% | 31.8% |
> | 0.1% context B | 75.1% | 85.8% | 34.5% | 51.3% | 51.5% | 19.3% |
>
> As reported in `Response to **Reviewer MQpJ W1**`, we observe similar trends on real-world reasoning tasks. We are also running **additional real-world experiments on reasoning tasks**, and promise to include these results in the rebuttal update before the rebuttal deadline.
>
> > W2 & Q2. Sensitivity of pass@128 to parser errors or formatting variations.
>
> Thank you for raising this point. The dependency-graph parser operates in a constrained setting: reasoning traces follow a fixed template family and the tokenizer is small (~2.2k tokens), which limits formatting variability. During development, we tracked multiple error categories and **iteratively hardened the parser** with normalization and fallback rules for recurring artifacts. On the gold set from the context test split 57k( solutions), the final parser achieves **100% graph construction coverage**, with **0 empty parses and 0 topological failures**. We will add this clarification in the appendix of the revised version.
>
> > Q1. Sensitivity to scale and architecture.
>
> Thanks for raising this important question. Regarding model scale, please see our response to `W1`, where we provide additional scaling evidence beyond the 100M setting. Overall, these results suggest that the main qualitative conclusions remain consistent as model size increases.
>
> Regarding base architecture, in preliminary experiments we also tested a LLaMA-style base model under the same controlled setup and did not observe meaningful qualitative differences in the main trends. In particular, the core findings about edge-of-competence RL and the role of pre-training exposure remained unchanged. Since these architecture comparisons are still preliminary, we have not included them in the main rebuttal for now, but if the reviewer is interested, we would be happy to add the corresponding results in the revision / appendix.
>
> > **Q3. Stability of the 1% contextual-exposure threshold.**
>
> GSM-Infinite contains three templates (teacher, zoo, movie). Across templates and multiple random seeds we observe very similar contextual-transfer behavior: templates differ mainly in surface world knowledge while the dependency graphs remain identical, and the transition to stable transfer around the ≈1% exposure regime remains consistent. Small variations in the definition of “atomic exposure” (e.g., different aggregation rules) also do not materially shift this transition. We will include the multi-template and multi-seed results in the appendix.
>
> However, this threshold should not be interpreted as a universal constant. Our experiments are conducted in a controlled setting where contexts share **identical reasoning paradigm** and differ mainly in surface semantics (world knowledge). In real-world domains with substantially different reasoning structures and environmental feedback, the exact threshold may vary. We will discuss this part in future work.
>
> > **Limitation. The paper does not adequately discuss limitations or negative societal impact.**
>
> We agree and will substantially revise the limitations/impact discussion. In particular, we will explicitly state the narrow empirical scope of the current paper: a synthetic arithmetic-like reasoning task, and a strict process-verified evaluation protocol. We will also make the risk of overgeneralization explicit, i.e., the current results should be interpreted as controlled evidence about stage interplay in this setting rather than as a blanket claim about all reasoning settings. More broadly, we will discuss the societal/alignment risk that reward-driven training can improve outcomes while still incentivizing invalid intermediate reasoning when the reward is not sufficiently process-aware.

---

> > ### Author Rebuttal · Reviewer_tWWJ · 2026-04-04
> >
> > Thank you for your detailed reply. My main concerns have been addressed, so I am happy to keep my positive score.

---

> > > ### Author Response · Authors · 2026-04-07
> > >
> > > We additionally scaled our experiments to a **500M Qwen2.5** model using the same training setup. The results remain consistent with our main conclusions:
> > >
> > > ### 1. RL on edge-of-competence data
> > >
> > > |Model|op2–10p@1|op2–10p@128|op11–14p@1|op11–14p@128|op15–20p@1|op15–20p@128|
> > > |-|-|-|-|-|-|-|
> > > |Base|**96.6%**|**99.8%**|47.4%|77.9%|14.6%|45.3%|
> > > |RL 7–10|90.6%|99.5%|40.3%|75.0%|16.0%|48.9%|
> > > |RL 9–12|90.4%|99.6%|53.0%|**88.6%**|23.7%|49.6%|
> > > |RL 11–14|90.3%|99.5%|**53.4%**|**88.6%**|**26.2%**|**53.2%**|
> > > |RL 17–20|90.3%|99.6%|40.6%|76.3%|16.3%|49.3%|
> > >
> > > **Takeaway 1.** RL is still most effective when trained on edge-of-competence data: RL on **op11–14** gives the best OOD extrapolation gains, including about +9 points in pass@128 on op15–20, while RL on easier or much harder data is less effective.
> > >
> > > ### 2. RL under different pre-training exposure to context B
> > >
> > > The table below reports results on **context B**.
> > >
> > > |Pre-train context B ratio|op2–10p@1|op2–10p@128|op11–14p@1|op11–14p@128|op15–20p@1|op15–20p@128|
> > > |-|-|-|-|-|-|-|
> > > |10%|**82.7%**|95.8%|**41.0%**|71.7%|**39.5%**|**54.0%**|
> > > |1%|82.1%|**98.0%**|39.4%|**76.0%**|31.8%|51.6%|
> > > |0.1%|75.1%|85.8%|34.5%|51.3%|19.3%|51.5%|
> > > |0|0|0|0|0|0|0|
> > >
> > > **Takeaway 2.** Contextual transfer still requires **minimal pre-training exposure**: with **0%** exposure to context B, RL fails completely, while even **sparse exposure (1–10%)** already enables strong transfer.
> > >
> > > ### 3. Mid-training + RL mixture
> > >
> > > |Model|op2–10p@1|op2–10p@128|op11–14p@1|op11–14p@128|op15–20p@1|op15–20p@128|
> > > |-|-|-|-|-|-|-|
> > > |Base|**96.6%**|99.8%|47.4%|77.9%|14.6%|45.3%|
> > > |Full RL|90.3%|99.5%|53.4%|88.6%|26.2%|53.2%|
> > > |80% RL|90.8%|99.5%|83.2%|98.1%|31.8%|66.3%|
> > > |60% RL|93.4%|99.8%|86.5%|99.1%|34.9%|**66.6%**|
> > > |40% RL|95.0%|**99.9%**|88.4%|98.7%|34.3%|62.6%|
> > > |20% RL|**96.2%**|99.8%|**89.9%**|**99.6%**|**36.4%**|63.5%|
> > >
> > > **Takeaway 3.** Mid-training + RL continues to outperform RL alone under the same compute budget. For the 500M model, the best mixture shifts somewhat compared with the smaller model, which is expected because the stronger base changes the optimal allocation between supervised bridging and RL exploration. Still, the overall conclusion remains unchanged: a hybrid mid-training + RL recipe is substantially better than pure RL.
> > >
> > > We also tested a harder setting where the mid-training and RL data are disjoint as suggested by `reviewer AVFA`: specifically, we replaced the mid-training data with op9–12 while keeping RL on op11–14.
> > >
> > > |Model|op2–10p@1|op2–10p@128|op11–14p@1|op11–14p@128|op15–20p@1|op15–20p@128|
> > > |-|-|-|-|-|-|-|
> > > |Base|**96.6%**|99.8%|47.4%|77.9%|14.6%|45.3%|
> > > |80% RL|91.5%|99.0%|71.5%|92.1%|17.6%|47.2%|
> > > |60% RL|93.6%|99.5%|76.0%|92.8%|18.7%|**49.3%**|
> > > |40% RL|94.4%|99.5%|**77.2%**|93.7%|**18.8%**|49.2%|
> > > |20% RL|95.7%|**99.7%**|76.7%|**94.3%**|16.3%|46.5%|
> > >
> > > The same qualitative trend still holds: combining mid-training and RL remains more effective than RL alone, even when the two stages are trained on disjoint data.
> > >
> > > ### 4. RL with process-aware outcome rewards
> > >
> > > |Model|Δop2–10p@1|Δop2–10p@128|Δop11–14p@1|Δop11–14p@128|Δop15–20p@1|Δop15–20p@128|ΔID|ΔOOD|ΔTotal|
> > > |-|-|-|-|-|-|-|-|-|-|
> > > |RL 7–10|-0.00|+0.10|+0.06|-0.44|+0.01|+0.00|+0.02|+0.04|+0.03|
> > > |RL 9–12|+0.04|-0.10|+0.19|-0.30|**+0.02**|**+0.30**|**+0.05**|+0.03|+0.04|
> > > |RL 11–14|**+0.06**|**+0.24**|**+0.52**|**+1.05**|-0.06|+0.03|+0.02|**+0.21**|**+0.12**|
> > >
> > > **Takeaway 4.** Process-aware rewards also bring additional gains, especially on OOD metrics.
> > >
> > > Overall, these 500M results show that our findings are **robust across scale**. We hope our experiments address your concerns. We have also added additional **real-world experiments** in our `response to Reviewer MQpJ`. If interested, please feel free to take a look there.

---

### Official Review · Reviewer_MQpJ · 2026-03-13

**Soundness:** 3
**Presentation:** 4
**Significance:** 3
**Originality:** 3
**Overall Recommendation:** 5
**Confidence:** 4

**Summary:**

This paper investigates a central question in LLM training: does reinforcement learning (RL) genuinely extend a model's reasoning capabilities beyond what pre-training instills, or does it merely sharpen existing skills? The authors construct a controlled experimental framework using synthetic math reasoning tasks (built on GSM-Infinite), manipulating pre-training, mid-training, and RL data distributions independently. They evaluate along two axes: extrapolative (depth) generalization and contextual (breadth) generalization, and draw four main conclusions: (1) RL produces true capability gains only at the model's "edge of competence"; (2) contextual generalization via RL requires at least minimal pre-training exposure to target contexts; (3) mid-training substantially boosts performance under fixed compute; and (4) process-level rewards reduce reward hacking.

**Compliance With Llm Reviewing Policy:**

Affirmed.

**Final Justification:**

The additional experiments conducted during the rebuttal further confirmed my initial judgement. This paper presents extensive experimental evidence of the contributions of different training regimes used in the standard recipe of modern LLM training.

I confirm my suggestion that this paper should be included in the conference.

The authors did not fully address my suggestion to better discuss the limitations of their work in light of the potential risks of unexpected LLM alignment. I advise them to include a brief discussion of this topic in the camera-ready version.

**Key Questions For Authors:**

- On Figure 5: The topological similarity analysis is computed over correct Context B solutions, which is the right methodological choice. However, did the authors perform the equivalent analysis over incorrect Context B solutions? This would help clarify whether generalization failures in the low-exposure regime stem from the model attempting novel but incorrect reasoning structures, or from rigid and incorrect replication of Context A patterns. The two failure modes have inherently different implications for interpreting the results

- Medium-RL is missing from Figure 6. Section 5's Task Setting explicitly describes five training configurations: Full Mid, Full RL, Light-RL  Medium-RL, and Heavy-RL. However, Figure 6 in the main body only shows four lines, Medium-RL is silently omitted. The Medium-RL results do appear in Figure 17 in Appendix K, but this omission is never acknowledged in the main text. Could you clarify this omission?

**Limitations:**

The authors include a brief impact statement but do not discuss the relationship between RL-based post-training and AI alignment. Despite the paper's focus on reasoning-oriented RL, this is a notable omission given that findings on reward hacking and process verification directly touch on a well-known alignment concern: that models optimizing for outcome-based rewards may learn to produce correct-looking answers through invalid reasoning chains. A brief acknowledgment of how RL-driven capability gains could amplify misaligned behaviors in more capable models would strengthen the impact statement.

**Strengths And Weaknesses:**

**Strengths**

- Controlled experimental design. The use of synthetic, contamination-free data with fully separable training distributions is the paper's strongest asset. This is a meaningful methodological advance over prior work that draws conclusions from opaque, real-world pre-training corpora. The ability to independently vary operation count ranges, context templates, and training phases gives the causal claims here substantially more credibility than most comparable work.

- Reconciliation of a genuine open debate. The finding that both camps in the "RL extends reasoning / RL only refines" debate are correct under different conditions, specifically, different difficulty regimes relative to the model's competence, is a useful and clean resolution. Figure 3 makes this point convincingly.

- Mid-training as underexplored lever. The compute-budget analysis in Section 5 is one of the more practically useful contributions. Showing that mid-training + light RL outperforms full RL on OOD-edge tasks, while heavy RL wins on OOD-hard tasks, gives practitioners actionable guidance that is not currently well-established in the literature.

- The paper is generally well-organized and the four main findings map cleanly onto sections. The Observation / Takeaway / Discussion / Practical Guidance structure is helpful for navigation, though it becomes slightly repetitive across sections. Figure 1 is a strong visual summary and the appendix is thorough.

- Extensive experimental coverage. The paper is remarkably thorough in its empirical scope. Beyond the four main sections, the appendix provides detailed training dynamics, reward curves, ablations across compute budgets, and data recipes for every experimental condition.

**Weaknesses**

- Scale is a significant concern. All experiments use a 100M-parameter model trained on synthetic data. The four takeaways are presented with fairly strong generality, but it is entirely unclear whether they hold at 1B, 7B, or 70B scale, or on naturalistic pre-training distributions. The synthetic setting, while controlled, may not capture emergent phenomena that arise in real-world training pipelines. This is the single largest limitation and should be discussed more prominently, ideally with at least a brief scaling experiment or a frank acknowledgment of where the results may not transfer.

- Novelty of individual claims is moderate. Several findings are in line with or extensions of prior work. The "edge of competence" framing echoes curriculum learning literature and is adjacent to prior findings in the literature. The claim that pre-training coverage is necessary for RL to generalize is similarly intuitive and has been noted informally in several papers. The paper's contribution is more in the rigorous, unified empirical validation than in the conceptual novelty of any single finding.

**Minor**

- Context A and Context B are never defined in the main body. These labels are load-bearing throughout Sections 3, 4, and the appendices, but their concrete definitions (A = animals–zoo, B = teachers–school, C = movie-festival) appear only in a footnote to Table 2 in Appendix J. The labels are first used in Section 4's Task Setting with no prior definition. A reader following the main text alone has no idea what A and B refer to. A single definitional sentence at the end of Section 2.1 would resolve this entirely.

- Line 182 lists "next-token" and "SFT objectives" as distinct options for mid-training, when SFT is technically also a next-token prediction objective, the distinction being the data used, not the loss function


----

Overall, I recommend accepting this paper. Despite the limitations noted above, particularly the restriction to 100M-parameter models and synthetic data, the paper makes a genuine and timely contribution. It addresses an actively debated question with a more controlled experimental framework than prior work, produces actionable practical guidance, and is thorough in its empirical coverage. The weaknesses are real but addressable through minor revisions, and they do not undermine the core contributions.

---

> ### Author Rebuttal · Authors · 2026-03-31
>
> Thanks for supporting the acceptance of our work. Please find our responses to your questions below.
>
> > W1. Scale is a significant concern. … The synthetic setting, while controlled, may not capture emergent phenomena that arise in real-world training pipelines.
>
> For the scaling experiments, we conducted 500M qwen 2.5 in our task. Please view the results on `response to reviewer tWWJ W1.`
>
> In addition, we conducted real-world experiments on **Qwen2.5-7B base** according to our findings. Difficulty buckets are constructed from **NuminaMath-CoT** (few-shot CoT) training corpus based on empirical pass rates of the base model:
>
> - **Easy:** pass@1 == 1
> - **Medium:** pass@32 ≠ 0
> - **Hard:** pass@128 ≠ 0
> We filtered 20k data for each bucket and ran GRPO with 80 steps. For evaluation, we split according to base model performance:
>
> - MATH-500: ID Task (pass@32 near 85%)
> - AIME2024 / 2025: OOD Task (low pass@32)
>
> The findings are similar to **Takeaway 1**:
>
> - ID task: pass@1 improves a lot, no substantial improvements on pass@k.
> - OOD task: both pass@1 and pass@k improve.
>
> #### MATH-500
>
> | Model / Setting | pass@1 | pass@2 | pass@4 | pass@8 | pass@16 | pass@32 |
> |-|-|-|-|-|-|- |
> | Baseline (Qwen2.5-7B) | 29.7 | 45.9 | 61.8 | 73.3 | 80.4 | **84.8** |
> | RL on Easy bucket | 65.2 | 70.2 | 74.2 | 77.0 | 79.0 | 80.6 |
> | RL on Medium bucket | 65.7 | 70.6 | 74.5 | 77.7 | 80.1 | 81.6 |
> | RL on Hard bucket | **65.9** | 71.2 | 75.2 | 78.1 | 80.1 | 81.4 |
>
> #### AIME 2024
>
> | Model / Setting | pass@1 | pass@2 | pass@4 | pass@8 | pass@16 | pass@32 |
> | --- | --- | --- | --- | --- | --- | --- |
> | Baseline (Qwen2.5-7B) | 3.1 | 5.6 | 9.3 | 14.4 | 21.3 | 30.0 |
> | RL on Easy bucket | 10.5 | 13.4 | 16.5 | 20.9 | 25.8 | 30.0 |
> | RL on Medium bucket | 10.9 | 14.3 | 17.8 | 21.7 | 25.7 | 30.0 |
> | RL on Hard bucket | **12.9** | 16.3 | 18.8 | 21.3 | 25.1 | 30.0 |
>
> #### AIME 2025
>
> | Model / Setting | pass@1 | pass@2 | pass@4 | pass@8 | pass@16 | pass@32 |
> | --- | --- | --- | --- | --- | --- | --- |
> | Baseline (Qwen2.5-7B) | 0.6 | 1.2 | 2.4 | 4.6 | 8.4 | 13.3 |
> | RL on Easy bucket | 4.8 | 8.5 | 13.7 | 19.8 | 26.7 | 33.3 |
> | RL on Medium bucket | 11.0 | 16.2 | 21.3 | 25.7 | 30.1 | 33.3 |
> | RL on Hard bucket | **11.3** | 18.5 | 27.8 | 38.2 | 48.2 | **60.0** |
>
> Also, we are currently running **additional real-world experiments on (i) cross-task generalization to coding and science domains (Takeaway 2 in our paper)** and **(ii) Mid-training & RL compute allocation (Takeaway 3).**  We will release the results before the rebuttal period ends.
>
> > **W2. Novelty of individual claims is moderate.**
>
> We appreciate this perspective. We agree that the main contribution is a **unified and tightly controlled empirical study** that disentangles the training stages under a  **process-verified setting**. In the revision, we will emphasize more clearly that the contribution lies in the systematic characterization of their interplay.
>
> > **Minor 1 & 2. Typo Issue**
>
> Thank you for pointing this out. We will add a one-sentence clarification in Section 2.1 that context refers to different semantic templates. Our intent was to distinguish the data regime, not the loss function: both pre-training and SFT use next-token prediction, while our mid-training simply narrows the data distribution toward the regime most relevant for RL.
>
> > **Q1. Equivalent analysis over incorrect Context B solutions**
>
> Thank you for this insightful suggestion. We examined the topological similarity distribution for incorrect Context B generations and found that most fall in the medium similarity range (0.4–0.7) rather than 0.8–1.0. This indicates that incorrect solutions exhibit partially similar but structurally varied reasoning, suggesting the model attempts novel reasoning paths that fail to generalize under low Context B exposure. This tendency also becomes more pronounced for harder tasks, supporting the view that failures arise from unstable or incomplete reasoning generalization, rather than simple pattern copying from Context A.
> | Context B | Difficulty | 0.1–0.3 | 0.4–0.7 | 0.8–1.0 |
> |:---------:|:--------:|--------:|--------:|--------:|
> | **0.1%** | 2–10 | 9.0 | 63.1 | 27.8 |
> | | 11–14 | 53.4 | 44.5 | 2.1 |
> | | 15–20 | 68.0 | 31.4 | 0.6 |
> | **1%** | 2–10 | 10.9 | 51.2 | 37.1 |
> | | 11–14 | 15.9 | 71.1 | 13.0 |
> | | 15–20 | 34.8 | 79.9 | 1.3 |
> | **10%** | 2–10 | 19.5 | 56.5 | 24.0 |
> | | 11–14 | 11.6 | 72.8 | 15.6 |
> | | 15–20 | 30.6 | 67.7 | 1.7 |
>
> > **Q2. Medium-RL is missing from Figure 6.**
>
> Thank you for pointing this out. Medium-RL does not exhibit a distinct qualitative pattern; its curve consistently lies between the Light-RL and Heavy-RL regimes and follows the same overall trend. To keep Figure 6 focused on the most informative contrasts, we omitted Medium-RL from the main figure for visual clarity and report the full results in **Figure 17 (Appendix K)**.
>
> We will add a clarification in the main text noting that Medium-RL results are provided in the appendix for completeness.

---

> > ### Author Rebuttal · Reviewer_MQpJ · 2026-04-01
> >
> > I thank the authors for their work on the rebuttal. I appreciate the effort put in scaling to a larger model and additional more realistic datasets. I confirm my score and the fact that this paper should be included in the conference.

---

> > > ### Author Response · Authors · 2026-04-08
> > >
> > > To further address the concern about real-world applicability, we conducted **additional experiments on Qwen2.5-7B Base**, and the conclusions remain consistent with our main claims.
> > >
> > > ### (1) RL on edge-of-competence data
> > > The corresponding results are reported above.
> > >
> > > **Takeaway 1.**
> > > Our main conclusion still holds at 7B scale: **RL works best on edge-of-competence data**. On in-domain tasks, it mainly improves **pass@1**, while on harder out-of-domain tasks, it improves both **pass@1** and **pass@k**.
> > >
> > > ---
> > >
> > > ### (2) Mid-training and RL compute allocation
> > >
> > > Using the same **edge-of-competence** data described above, we further study how to allocate a **fixed training budget of 100 steps** between mid-training and RL. We use the same compute conversion between the two stages and compare the following allocations: **Full Mid-training**, **80/20**, **50/50**, **20/80**, and **Full RL**, where the ratio denotes the fraction of total compute assigned to **mid-training vs. RL**.
> > > ||MATH-500|||| AIME 2024 |||| AIME 2025 ||||
> > > |---|-:|-:|-:|-:|-:|-:|-:|-:|-:|-:|-:|-:|
> > > |Setting|p@1|p@4|p@8|p@32|p@1|p@4|p@8|p@32|p@1|p@4|p@8|p@32|
> > > |Base|29.7|61.8|73.3|**84.8**|3.1|9.3|14.4|30.0|0.6|2.4|4.6|13.3|
> > > |`Full mid`|**65.4**|73.9|76.9|80.2|**14.2**|20.0|23.1|30.0|**9.8**|17.5|22.9|33.3|
> > > |`80/20`|58.7|73.1|77.4|82.6|7.9|15.8|21.2|33.3|7.1|16.2|20.6|33.3|
> > > |`50/50`|58.7|73.4|77.9|83.6|8.0|13.1|15.4|20.0|8.3|19.9|25.6|33.3|
> > > |`20/80`|65.2|**75.9**|**78.8**|82.2|12.7|**20.7**|**25.7**|**40.0**|9.0|**21.1**|**27.1**|**40.0**|
> > > |`Full RL`|64.9|73.9|77.3|82.0|10.7|20.6|25.3|36.7|6.0|15.4|21.0|26.7|
> > >
> > > **Takeaway 2.**
> > > With **fixed compute**, **mid-training** is better for improving **pass@1**, while allocating more budget to **RL** yields stronger higher-`k` performance, especially on harder benchmarks. This again shows that the two stages play **complementary roles**.
> > >
> > > ---
> > >
> > > ### (3) Cross-task generalization
> > >
> > > We also explored **cross-task generalization** from **math** to **code** and **science** for **Qwen2.5-7B Base**.
> > >
> > > For **code**, we evaluate on **LiveCodeBench v4** and **LiveCodeBench v6**. For **science**, we evaluate on **GPQA Diamond**.
> > >
> > > Since our budget does not allow pre-training from scratch, we start from the base model and inject target-domain knowledge via **pre-RL mid-training / SFT**. Concretely, before RL, we inject either **code** or **science** data into the training mixture. We consider two small injection ratios—**1%** and **5%**—for the target domain, with the remaining data drawn from the math **edge-of-competence** set described above.
> > >
> > > For RL, we then rebuild the training mixture as a **balanced 50/50 composition**, and compare settings such as **code+math** and **science+math**.
> > >
> > > #### Code
> > >
> > > |Setting|LCBv4 p@1|LCBv4 p@4|LCBv4 p@8|LCBv6 p@1|LCBv6 p@4|LCBv6 p@8|
> > > |---|-:|-:|-:|-:|-:|-:|
> > > |Base|0.6|1.9|3.0|2.1|6.9|10.3|
> > > |`1% Code`|||||||
> > > |pre-RL|1.0|3.3|5.0|2.4|7.0|9.7|
> > > |RL|4.8|10.5|13.9|9.7|15.4|17.7|
> > > |`5% Code`|||||||
> > > |pre-RL|0.7|2.5|4.0|2.4|7.5|11.4|
> > > |RL|**6.6**|**12.1**|**14.9**|**10.4**|**17.4**|**20.0**|
> > >
> > > #### Science
> > >
> > > |Setting|GPQA p@1|GPQA p@4|GPQA p@8|GPQA p@16|
> > > |---|-:|-:|-:|-:|
> > > |Base|27.8|69.4|**88.5**|98.0|
> > > |`1% Science`|||||
> > > |pre-RL|29.7|61.7|75.7|91.9|
> > > |RL|29.0|67.7|84.4|96.0|
> > > |`5% Science`|||||
> > > |pre-RL|31.1|67.3|84.4|**98.5**|
> > > |RL|**32.4**|**70.0**|85.6|98.0|
> > >
> > > **Takeaway 3.**
> > > A small amount of target-domain injection before RL enables transfer to **code** and **science**, with especially clear gains on **code**.
> > >
> > >
> > > We summarize these additional real-world results as follows:
> > >
> > > 1. **Edge-of-competence data remains the most effective RL regime at 7B scale.**
> > > 2. **Under a fixed compute budget, mid-training and RL should be allocated differently depending on the target metric:** mid-training is stronger for **pass@1**, while RL is more effective for improving higher-`k` performance.
> > > 3. **Our findings extend beyond math**, showing promising transfer to **code** and **science** with lightweight target-domain injection.
> > >
> > > Overall, these new experiments substantially strengthen the paper’s external validity: the same three core conclusions—about **edge-of-competence RL**, **mid-training/RL complementarity**, and **cross-domain transfer with lightweight domain seeding**—all continue to hold in a realistic **Qwen2.5-7B** setting. We hope these additional real-world results help address the reviewer’s concerns.
> > >
> > > ----
> > >
> > > We have also added additional **scaling experiments** in our `response to Reviewer tWWJ`, extending the controlled setup to a **500M Qwen2.5 model**, where we observe trends consistent with our main conclusions.
> > >
> > > If interested, please feel free to take a look there.

---

### Decision · Program_Chairs · 2026-04-30

**Decision:**

Accept (spotlight)

**Comment:**

This paper studies the interplay between pre-training, mid-training, and reinforcement learning in reasoning language models through a carefully controlled experimental framework. Using synthetic reasoning tasks with explicit structure and process-verifiable traces, the paper aims to isolate the causal contributions of different training stages and clarify a central debate in the literature: whether RL produces genuine capability gains or mainly sharpens existing abilities. The paper’s main claims are that RL is most effective at the model’s edge of competence, that contextual transfer requires minimal but non-zero prior exposure, that mid-training plays a central role under fixed compute budgets, and that process-level rewards improve reasoning fidelity and reduce reward hacking.

The reviewers were broadly very positive about the submission. A major strength repeatedly highlighted is the unusually controlled experimental design, which gives the paper a level of causal interpretability that is often missing in work on modern LLM training pipelines. Reviewers also found the paper thorough, well organized, and practically useful, especially in its analysis of compute allocation between mid-training and RL, and in its reconciliation of competing views on the role of RL in reasoning improvement. Several reviewers explicitly noted that, even when some of the high-level conclusions are directionally consistent with prior intuition, the contribution here lies in the systematic and unified empirical validation under a carefully designed process-verified setting.

The main concern in the initial reviews was external validity. In particular, reviewers questioned how far conclusions drawn from a 100M model trained on synthetic reasoning tasks could be generalized to larger models or more realistic settings. The rebuttal substantially strengthened the paper on exactly this point. The authors added scaling evidence on a 500M Qwen2.5 model and also reported real-world experiments on Qwen2.5-7B across math, code, and science settings, with trends that were consistent with the paper’s main claims. These additions appear to have resolved the main skepticism. Reviewers who raised these concerns explicitly stated that their questions had been addressed, and one reviewer raised their score after rebuttal. The remaining issues are mostly about clearly stating scope, limitations, and the risks of overgeneralization, rather than about the validity of the core empirical findings.

Overall, I find this to be a strong and timely empirical study. The paper does not claim a new algorithmic breakthrough, but it makes a meaningful contribution by providing a controlled framework for understanding how different training stages interact in reasoning models, and by extracting concrete lessons that appear to transfer beyond the smallest synthetic setting. Given the strength of the reviewer consensus after discussion, the thoroughness of the empirical study, and the substantial rebuttal improvements, I recommend **accept**. In the final version, the authors should more explicitly emphasize the paper’s scope, clarify that several conclusions are empirical characterizations rather than universal laws, and strengthen the limitations and societal impact discussion accordingly.